# Renal CD169[++] resident macrophages are crucial for protection against acute systemic candidiasis

Yi Juan Teo[1] , See Liang Ng[1], Keng Wai Mak[1], Yolanda Aphrilia Setiagani[1], Qi Chen[1] , Sajith Kumar Nair[1], Jianpeng Sheng[2], Christiane Ruedl[1]

Disseminated candidiasis remains as the most common hospital-acquired bloodstream fungal infection with up to 40% mortality rate despite the advancement of medical and hygienic practices. While it is well established that this infection heavily relies on the innate immune response for host survival, much less is known for the protective role elicited by the tissue-resident macrophage (TRM) subsets in the kidney, the prime organ for *Candida* persistence. Here, we describe a unique CD169[++] TRM subset that controls *Candida* growth and inflammation during acute systemic candidiasis. Their absence causes severe fungal-mediated renal pathology. CD169[++] TRMs, without being actively involved in direct fungal clearance, increase host resistance by promoting IFN-γ release and neutrophil ROS activity.

## Introduction

Systemic candidiasis is the fourth common bloodstream nosocomial infection that was estimated to affect more than 250,000 intensive care unit patients every year, despite the administration of hygienic practices in the hospitals (Delaloye & Calandra, 2014; Kullberg & Arendrup, 2015). Although current treatment of this infection mainly uses antifungal drugs, the mortality rate among patients remains alarmingly high (40–60%) (Delaloye & Calandra, 2014; Bassetti et al, 2018; Lamoth et al, 2018). Yet, no vaccines are clinically available to date. With the increasing population of inpatients suffering from chronic illnesses and the emerging cases of antifungal drugs resistance, understanding the host immunity against such infection will be important in developing immunotherapy to improve or complement the current antifungal interventions (Armstrong-James et al, 2017; Desai et al, 2017; Bassetti et al, 2018).

Widely known as the central players in invasive antifungal immunity, phagocytes, in particular, polymorphonuclear phagocytes (neutrophils) use various mechanisms in controlling fungal infections, such as phagocytosis, the release of reactive oxygen species (ROS) and microbicidal proteins, and NETosis formation (Borregaard, 2010; Mantovani et al, 2011). The pivotal role of neutrophils in *Candida* immunity is further underlined by the clinical association of neutropenia and neutrophil defects as the predisposing factors toward systemic candidiasis (Lehrer & Cline, 1969; Wisplinghoff et al, 2004; Pfaller & Diekema, 2007; Yapar, 2014). However, much less is known in the roles of different mononuclear phagocytes in this infection in vivo, in part owing to the lack of available tools in delineating the different macrophages and dendritic cells subsets in the kidneys, which are the main target organs of systemic candidiasis (Schraml et al, 2013; Gottschalk & Kurts, 2015).

The importance of macrophages in invasive *Candida* immunity has, to the best of our knowledge, first been strongly demonstrated by Lionakis et al (2013). Taking advantage of CX3CR1[gfp/gfp] mice, wherein their numbers of renal macrophage population are greatly reduced, Lionakis et al (2013) showed that renal resident macrophages are the crucial first-line defenders against the assault of *C. albicans* (Lionakis et al, 2013). Also, these macrophages seemed to be involved in regulating neutrophil recruitment to the kidneys during *Candida* infection (Kanayama et al, 2015). Besides promoting fungal clearance, macrophages were reported to play a role in renal tissue repair (Tran et al, 2015).

CD169, also known as Sialoadhesin or sialic acid–binding immunoglobulin-like lectin 1 (Siglec-1), has previously been reported to be a specific marker that identifies tissue-resident macrophages (TRMs) in various peripheral organs such as lungs, spleen, liver, and kidneys (Purnama et al, 2014; Karasawa et al, 2015; Gupta et al, 2016; Svedova et al, 2017). Interestingly, renal CD169[+] macrophages have been associated with immunoregulation, either toward the progression of immunopathology or immune resolution, depending on the disease/injury models (Chavez-Galan et al, 2015). However, little is known about the in vivo functional role of CD169[+] macrophages in systemic *Candida* immunity.

Here, we show that renal CD169[++] macrophages are important immune regulators in acute systemic *Candida* infection. Absence of CD169[++] macrophages diminishes IFNγ response and neutrophil ROS production in the kidneys. As a result, mice that lack CD169[++] macrophages succumb to a low-dose *Candida* infection, exhibiting exceedingly high fungal burden and severe renal immunopathology.

[1]School of Biological Sciences, Nanyang Technological University, Singapore, Singapore  [2]Department of Hepatobiliary and Pancreatic Surgery, The First Affiliated Hospital, Zhejiang University School of Medicine, Hangzhou, China

Correspondence: yijuan2010@gmail.com; ruedl@ntu.edu.sg

# Results

## CD169[++] macrophages are a subpopulation of renal TRMs

To investigate renal TRMs, we exploited a CD169-DTR transgenic mouse model that specifically ablates TRMs upon diptheria toxin (DT) treatment because of their CD169 expressions (Purnama et al, 2014; Gupta et al, 2016; Chen & Ruedl, 2020). Besides CD169, high expression level of F4/80 and intermediate level of CD11b was used in our flow cytometry analysis to distinguish TRMs from other macrophage subpopulations (Sheng et al, 2015) (Fig 1A). Interestingly, only a partial population of renal F4/80[hi] CD11b[int] macrophages was ablated in our DT treated CD169-DTR transgenic mice (Fig 1A and B), suggesting heterogeneity of F4/80[hi] CD11b[int] macrophage population in the kidney. Parallel with our observation, Karasawa et al (2015) also pointed out that only a subset of the renal TRMs express CD169 (Karasawa et al, 2015). In particular, they showed that CD169[++] TRMs mainly localize in the renal medullary region, corroborating with our immunofluorescence stainings (Fig 1C).

Here, we noted that ablated CD169[++] TRMs express moderately lower levels of F4/80 and CD11b (F4/80[++] CD11b[+]), whereas the unablated TRMs express comparatively higher levels of F4/80 and CD11b (F4/80[+++] CD11b[++]). However, these two TRM populations are indistinguishable in WT mice (Fig 1A). Hence, based on the ablated and unablated TRMs, we broadly subcategorized them into Fraction I (Fr I) (CD169[++] F4/80[++] CD11b[+]) and Fraction II (Fr II) (CD169[+] F4/80[+++] CD11b[++]) populations (Fig 1A and B). Noteworthy, the noticeable, yet insignificant reduction of the Fr II population in CD169-DTR mice also indicates that some of the Fr II population expresses CD169. The TRM ablation profile in CD169-DTR mouse is consistent with the CD169 transcript expression, wherein Fr I population express significantly higher level of CD169 as compared with Fr II population (Fig 1D). Correspondingly, higher levels of human heparin-binding EGF-like growth factor (HB-EGF), the receptor for DT, were observed in Fr I when compared with Fr II TRMs, explaining their susceptibility to DT treatment (Fig 1E).

## Absence of renal CD169[++] macrophages greatly compromised the host's resistance against disseminated candidiasis

To assess the importance of CD169[++] macrophages, CD169-DTR and WT mice were i.v. challenged with a low dose of $5 \times 10^4$ cfu *C. albicans*, and their survival was monitored for 18 d. During the course of infection, the mice were continuously treated with DT to ensure continual ablation of Fr I population in CD169-DTR mice (Fig S1). A partial reduction of Fr I macrophages was also observed in infected WT mice at day 3 postinfection (p.i.) (Fig 1F), which could be a result of macrophage necroptosis, an event that has been reported to occur during pathogenic infections (Lai et al, 2018; Cao et al, 2019). Conversely, the full ablation of Fr I macrophages in CD169-DTR mice was predominantly due to apoptosis induced by DT treatment. Nevertheless, the number of Fr I macrophages in CD169-DTR mice were consistently and substantially lower than those in the WT mice throughout the course of infection. On the other hand, the total numbers of Fr II macrophages were comparable between infected WT and CD169-DTR mice (Fig 1F, right panel). Consequently, CD169-DTR mice, which lack of Fr I TRMs, were distinctively more vulnerable than WT mice when challenged with systemic *Candida* infection (Fig 1G).

CX3CR1[gfp/gfp] mice have previously been used to interrogate the function of kidney TRMs due to its loss of total renal F4/80[+] CD11b[+] population (Lionakis et al, 2013). The comparison between CX3CR1[gfp/gfp] and CD169-DTR mice, therefore, allows us to understand the functional consequences between the partial and total loss of renal TRMs population. To achieve this, we infected CD169-DTR, CX3CR1[gfp/gfp] and WT mice with $5 \times 10^4$ cfu *C. albicans*. Similar to what was previously reported, CX3CR1[gfp/gfp] mice displayed a complete absence of renal TRMs (both Fr I and II populations), whereas CD169-DTR mice exhibited a more distinctive loss of Fr I population (Fig S2A and B). Correspondingly, lack of both Fr I and II macrophages rendered CX3CR1[gfp/gfp] mice to be very susceptible toward systemic candidiasis (Fig S2C), with an exceedingly high renal fungal burden as early as day 1 p.i. when compared with infected WT and CD169-DTR mice (Fig S2D). In contrast, CD169-DTR mice displayed lower mortality than CX3CR1[gfp/gfp] mice (Figs 1G and S2C), and their renal fungal burden was only significantly higher than the infected WT mice at day 10 (Fig 2A and B).

Since DT-treated CD169-DTR and CX3CR1[gfp/gfp] mice lack TRMs in other organs (Hochheiser et al, 2013; Gupta et al, 2016; Lee et al, 2018), we assessed the fungal loads in multiple peripheral organs, that is, kidney, brain, heart, lung, liver, and spleen, in infected WT, CD169-DTR, and CX3CR1[gfp/gfp] mice at day 1 p.i. Intriguingly, in both CD169-DTR and CX3CR1[gfp/gfp] mice, kidneys were the only organs displaying distinctly higher fungal burden than the rest of the organs investigated (Fig S2D). The elevated amount of *Candida* burden in CX3CR1[gfp/gfp] kidneys at day 1 reaffirms the findings reported by Lionakis et al (2013), wherein renal TRMs are vital for renal first-line defense against systemic *Candida* infection (Lionakis et al, 2013).

On the other hand, the renal fungal burden in CD169-DTR mice was significantly lower than those in CX3CR1[gfp/gfp] mice but not distinctly higher than the infected WT kidneys at day 1 p.i. (Fig S2D). Because Fr II population remains relatively intact in CD169-DTR mice (Fig S2A and B), we questioned whether the increased resistance observed in CD169-DTR mice, when compared with CX3CR1[gfp/gfp] mice, was mainly due to the remaining unablated renal Fr II macrophages that function to impede most *C. albicans* from invading into the renal tubules, an area where very little immune cells were detected during the steady state. To this end, we monitored the fungal burden of various organs in WT and CD169-DTR mice at day 1, 3, 6, and 10 p.i. (Fig 2). Here, our data showed kidney-restricted dependency of CD169[++] macrophages in fungal clearance as all the organs investigated, except kidneys, in CD169-DTR mice were cleared of *C. albicans* by day 10 p.i. (Fig 2A).

Interestingly, despite the loss of Fr I subset, fungal burden in CD169-DTR mice accrued similarly as the infected WT mice (day 0–6) (Fig 2A and B), suggesting minimal roles of renal CD169[++] macrophages in eliciting first-line defense against this infection. The disparity in kidney fungal burden between CD169-DTR and WT mice was only detected at day 10 p.i., wherein the fungal load of CD169-depleted kidneys continued to escalate, whereas those in WT mice seemed to be contained or cleared (Fig 2B). Therefore, our data demonstrate the indispensable role of CD169[++] macrophages in renal *Candida* immunity but likely with little contribution to the renal first-line defense against such infection.

Strikingly, since CX3CR1[gfp/gfp] mice lacked both Fr I and Fr II macrophages and displayed enhanced susceptibility as early as day 1 p.i. (Fig S2), we hypothesized that Fr II CD169[+] macrophages are predominantly responsible for the critical innate mechanism of early renal fungal control.

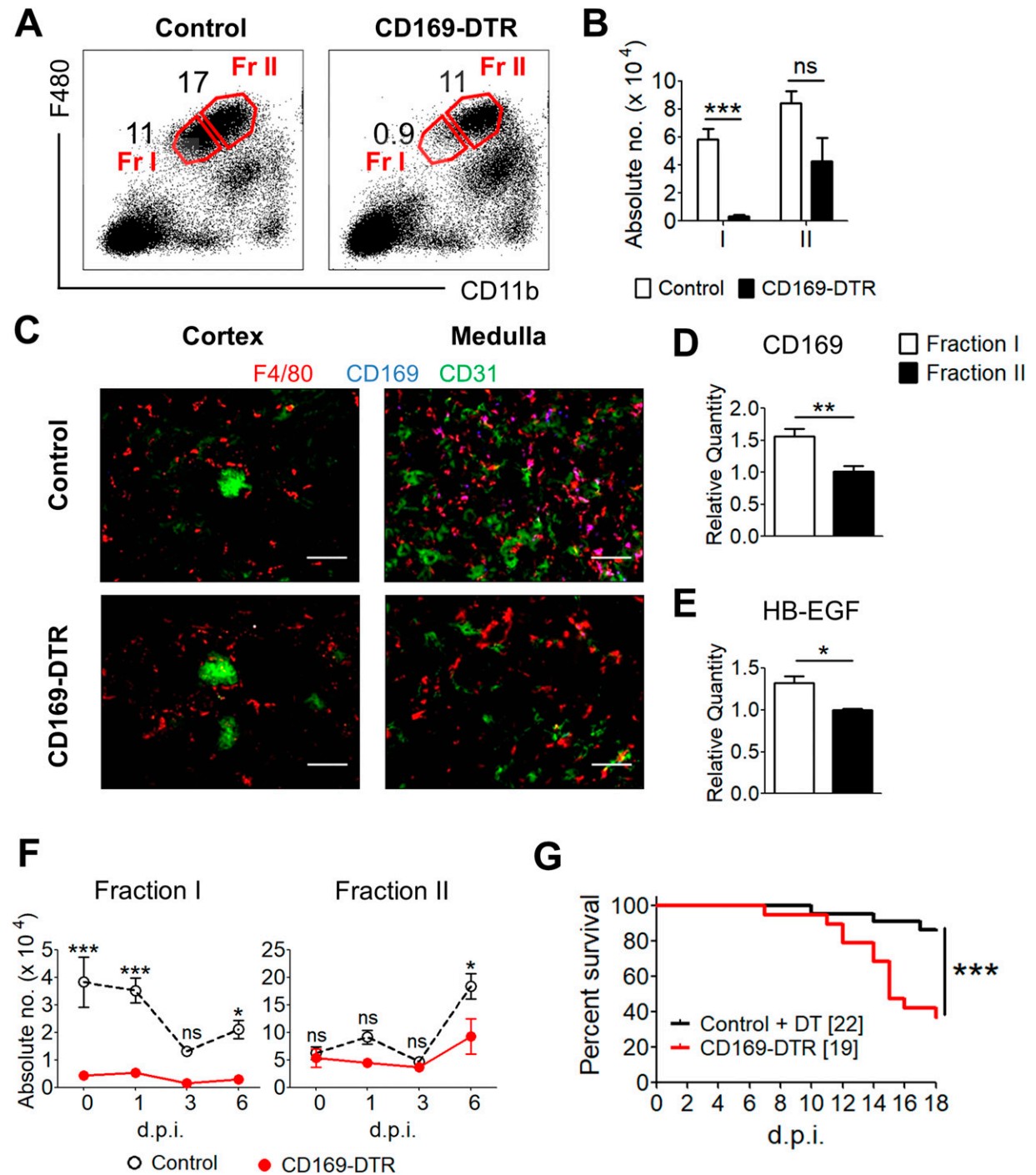

**Figure 1. Tissue-resident F4/80$^{++}$ CD11b$^{+}$ macrophages are efficiently ablated in the kidneys and are indispensable for systemic *Candida* immunity.**
**(A)** Representative flow cytometry profile indicating two subpopulations of kidney tissue-resident macrophages (TRMs), Fr I—F4/80$^{++}$ CD11b$^{+}$—and Fr II—F4/80$^{+++}$ CD11b$^{++}$— from control and CD169-DTR mice. **(B)** The total number of Fr I and II TRMs in the kidneys of control and diptheria toxin (DT)–treated CD169-DTR mice. *t* test (two-tailed). **(C)** Fluorescent immunostaining of control and CD169-DTR renal cortical and medullary cryosections with antibodies against F4/80 (red), CD169 (blue), and CD31 (green). Magnification, 200×. **(D, E)** Single-cell suspension of kidney was prepared and sorted for Fr I and II TRMs. **(D, E)** These subpopulations were then assessed for the relative transcript levels of (D) CD169 and (E) HB-EGF. *t* test (two-tailed). **(F)** Total number of Fr I and II TRMs in the kidneys of control and DT-treated CD169-DTR mice on day 0, 1, 3, and 6 after infection. Two-way ANOVA (Bonferroni posttests). **(G)** Representative survival curve of control and CD169-DTR mice after challenged i.v. with 5 × 10$^{4}$ *Candida albicans*. Log-rank (Mantel–Cox) test. Data are shown as mean ± SEM. *$P$ < 0.05. ***$P$ < 0.001. ns, not significant. Data represent two (A, B, C, D, F) (*n* = 3–4) or three (E) independent experiments.

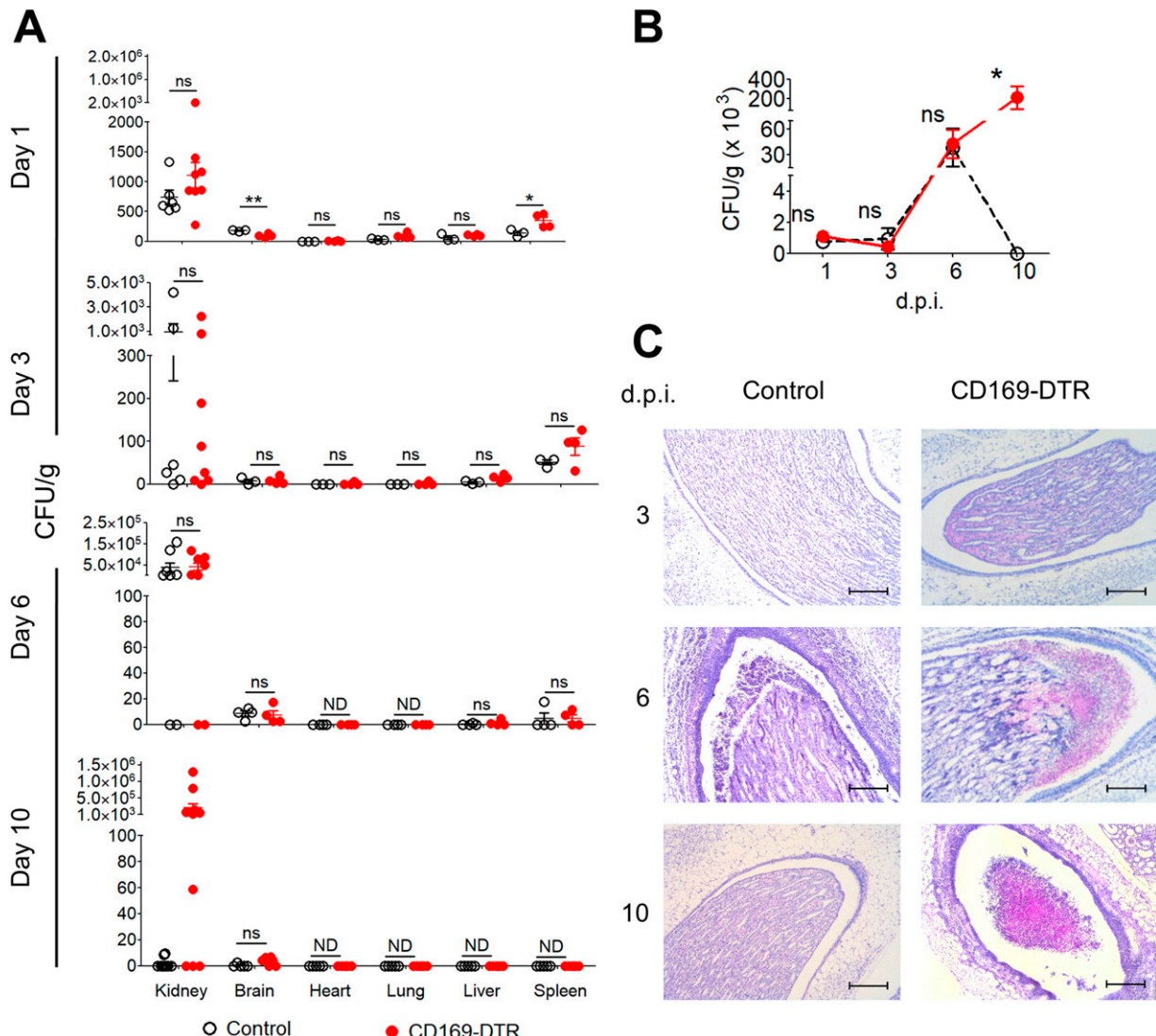

**Figure 2. Susceptibility of CD169-DTR mice is associated with ineffective renal *Candida* clearance.**
**(A)** Fungal burdens in the kidney, brain, heart, lung, liver, and spleen of control and CD169-DTR mice on day 1, 3, 6, and 10 after infection. *t* test (two-tailed). **(B)** Fungal burden in the kidneys of control and CD169-DTR mice on day 1, 3, 6, and 10 after infection. Two-way ANOVA (Bonferroni posttests). **(C)** Representative micrographs of periodic acid–Schiff–stained paraffin sections of infected kidneys of control and CD169-DTR mice on day 3, 6, and 10 after infection. Magnification, 100×. Scale bar, 200 µm. Data are shown as mean ± SEM. *$P < 0.05$. **$P < 0.01$. ND, not detectable. ns, not significant. Data represent two ($n = 3$) independent experiments.

## CD169-DTR mice suffered from irreversible, progressive renal damage during *Candida* infection

Because kidneys were the only organs where *C. albicans* accumulated during the infection (Fig 2A), we next performed histopathologic examination of periodic acid–Schiff (PAS) (Fig 2C)– and H&E (Figs 3A and S3)–stained sections obtained from infected WT and CD169-DTR kidneys at day 3, 6, and 10 p.i. At day 3, signs of renal damage and hemorrhages were observed in CD169-DTR kidneys (Fig S3A). At day 6, infected WT kidneys displayed minor signs of tubular and endothelial damage, whereas infected CD169-DTR kidneys showed more severe hemorrhages and tubular necrosis (Fig S3B). In both infected WT and CD169-DTR kidneys, most *C. albicans* appeared to be accruing in the renal pelvis (Fig 2C). Noteworthy, there was an accumulation of leukocytes in situ in both WT and

CD169-DTR kidneys, suggesting that the uncontrolled *C. albicans* growth in CD169-DTR kidneys is not due to the lack of leukocytes recruitment (Figs 2C and 4A).

At day 10, the structural integrity of CD169-DTR kidneys deteriorated, accompanied by the appearance of fibrotic tissues (Figs 3A and S3C). Specifically, enlargement of the Bowman's space and prominent shedding of tubular epithelial cells and clusters of leukocytes within the lumen of some renal tubules of CD169-DTR kidneys were clearly visible (Fig 3A). In contrast, by day 10, most of the renal regions in WT kidneys appeared to remain intact, indicating recovery from early infection (Figs 3A and S3C).

Parallel to the kidney histological analyses, we assessed the level of kidney injury molecule-1 (Kim-1) in infected WT and CD169-DTR kidney, a type-1 transmembrane protein where its expression is augmented only upon injury in the renal tubules (Ichimura et al,

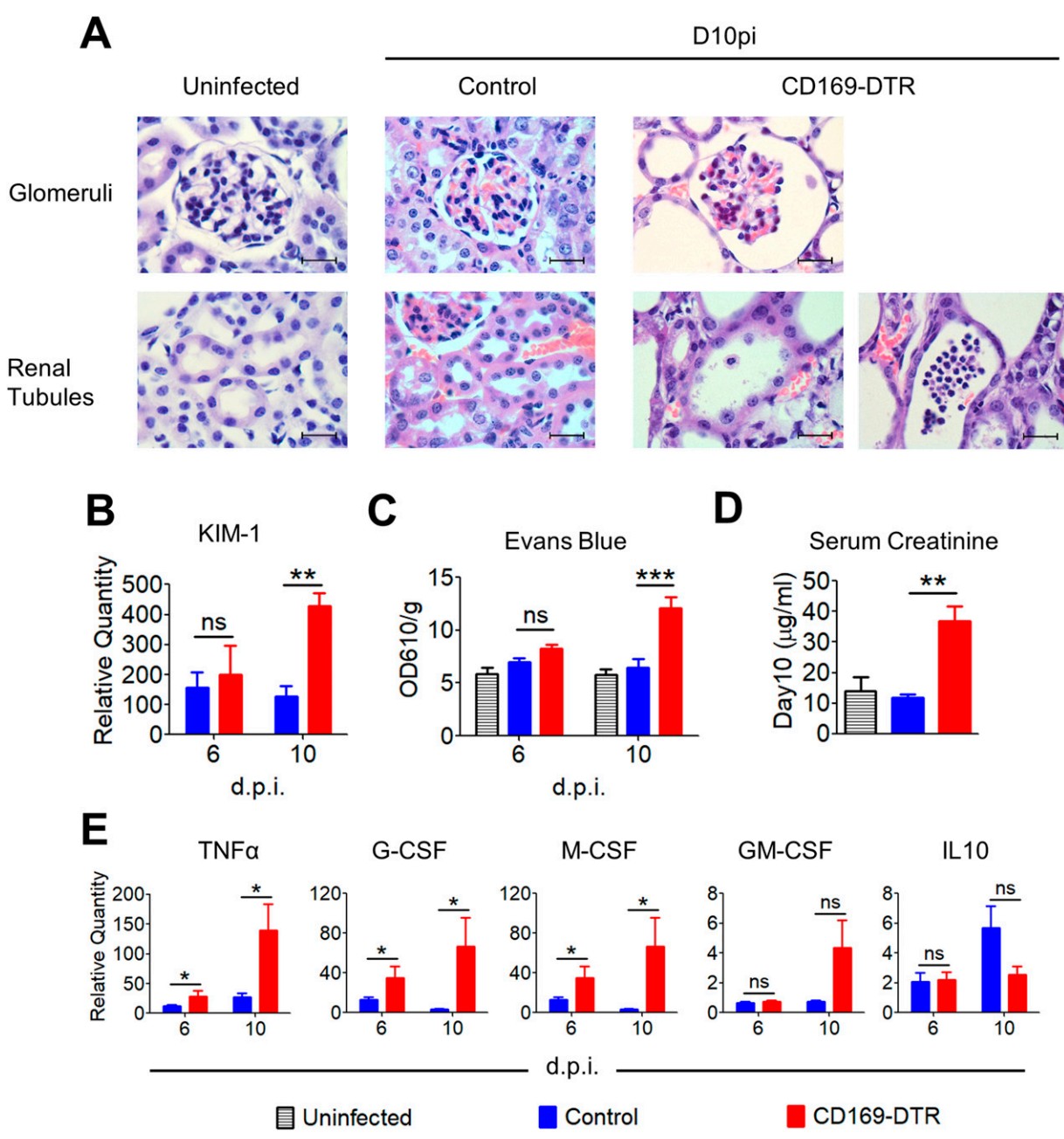

**Figure 3. CD169-DTR mice suffered from progressive kidney damage, inflammation, and compromised renal function.**
**(A)** Representative micrographs of H&E–stained paraffin sections of uninfected (day 0) and infected (day 10) kidneys of control and CD169-DTR mice. Magnification, 1,000×. Scale bar, 20 $\mu m$. **(B)** Relative mRNA expression levels of Kim-1 in infected kidneys, measured by quantitative RT-PCR. Data are normalized to $\beta$-actin. $t$ test (two-tailed). **(C)** Vascular permeability was measured by Evans blue extravasation, calculated from absorbance readings at $OD_{610}$. Data are normalized to organ weight (g). One-way ANOVA (Bonferroni's multiple comparisons test). **(D)** Serum creatinine (micrograms per milliliter), measured on day 10 after infection. $t$ test (two-tailed). **(E)** Relative expression levels of TNF$\alpha$, G-CSF, M-CSF, GM-CSF, and IL10 in kidneys, measured by quantitative RT-PCR. Data are normalized to $\beta$-actin. $t$ test (two-tailed). Data are shown as mean ± SEM. *$P < 0.05$. **$P < 0.01$. ***$P < 0.001$. ns, not significant. Data represent two ($n = 3$) independent experiments.

1998; Han et al, 2002; Bonventre, 2009). Our data showed that the expression of Kim-1 in the kidneys of infected CD169-DTR mice was drastically higher than in the infected WT at day 10 p.i. (Fig 3B). In addition, the integrity of the endothelial lining in the kidneys of CD169-DTR mice was significantly debilitated, as evidenced by higher amount of Evan's Blue retained in the CD169-DTR kidneys (Fig 3C).

CD169+ macrophages have previously been shown to be anti-inflammatory, wherein absence of these macrophages led to over-inflammation in bacterial infection or ischemia-reperfusion injury (Karasawa et al, 2015; Svedova et al, 2017). Consistent with these models, kidneys of CD169-DTR mice were associated with higher levels of inflammation (i.e., TNF-$\alpha$, G-CSF, and M-CSF) and renal function was found to be dramatically compromised when

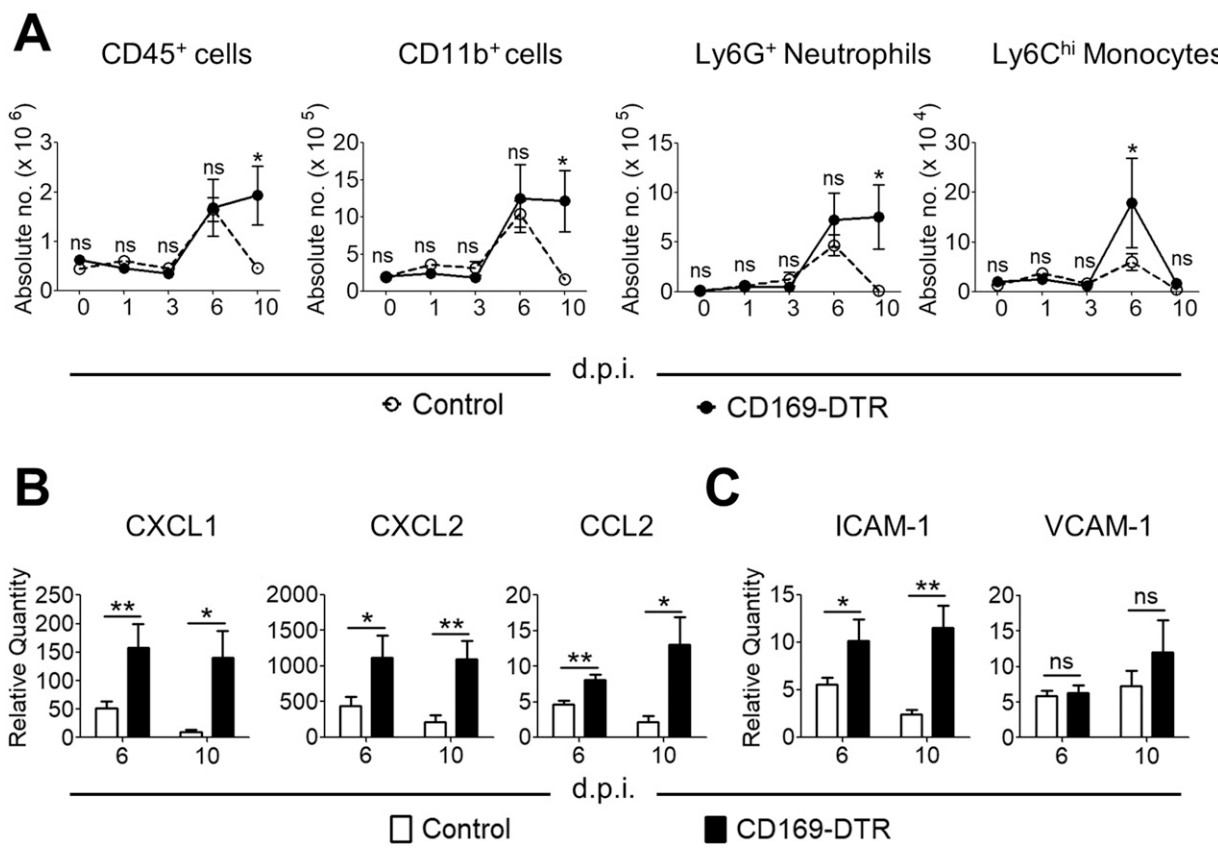

**Figure 4. Cellular infiltration is not impaired in the kidney of CD169-DTR mice.**
**(A)** Absolute number of CD45[+], myeloid (CD11b[+]), neutrophils (Ly6G[hi]CD11b[+]), and monocytes (Ly6C[hi]CD11b[+]) on day 0, 1, 3, 6, and 10 after infection. Two-way ANOVA (Bonferroni posttests). **(B)** Relative expression of transcripts for CXCL1, CXCL2, and CCL2, measured by quantitative RT-PCR. Data are normalized to β-actin. t test (two-tailed). **(C)** Relative expression of ICAM-1 and VCAM-1 mRNA in the kidneys, measured by quantitative RT-PCR. t test (two-tailed). Data are normalized to β-actin. Data are shown as mean ± SEM. *$P < 0.05$. **$P < 0.01$. ***$P < 0.001$. ns, not significant. Data represent two ($n = 3$) independent experiments.

compared with the infected WT kidneys (Fig 3D and E). Hence, highly inflamed kidneys at day 6 p.i. could contribute to the increased renal damage (tubular necrosis and hemorrhages) in CD169-DTR mice despite showing similar fungal burden as the infected WT mice (Fig 2B).

Taken together, we showed that kidneys of CD169-DTR mice were incapable of clearing *C. albicans* infection, which irreversibly aggravated renal damage as the disease progressed.

### Absence of renal CD169[++] macrophages did not impair the recruitment of effector cells during *Candida* infection

Because the inefficient clearance of *Candida* in CD169-DTR kidneys could be due to the lack of effector cells recruitment, we monitored neutrophilic and monocytic infiltration in the kidneys of WT and CD169-DTR mice at day 0, 1, 3, 6, and 10 p.i. Similar to our histological observations (Fig 2C), absence of renal CD169[++] Fr I macrophages did not affect the recruitment of leukocytes during the infection (Fig 4A). Instead, CD169-DTR kidneys were associated with higher expression of neutrophil- and monocyte-attracting chemokines (e.g., CXCL1, CXCL2, and CCL2) and cell adhesion molecules (e.g., ICAM-1) (Fig 4B and C). The increased level of chemokines in CD169-DTR kidney, potentially due to the unrestrained growth of *C. albicans*,

correlates with elevated amount of immune cells infiltration from day 6 to 10 p.i. In contrast, there was little sign of *C. albicans* in WT kidneys, and immune cells were largely cleared off at day 10 p.i. (Figs 2B and 4A). Thus, our data suggest that renal CD169[++] macrophages were not the major players in recruiting effector cells, in particular neutrophils, during *Candida* infection.

### Neutrophils in CD169-DTR kidneys generate lower ROS production

Next, we proceeded to interrogate whether the lack of CD169[++] macrophages adversely impact host candidacidal response in the kidneys. To this end, we assessed the amount of ROS-producing neutrophils and monocytes in infected WT and CD169-DTR kidneys at day 6 p.i.—the day when both WT and CD169-DTR kidneys displayed similar fungal burden and cellular infiltrations (Figs 2B and 4A). Interestingly, we observed lower amount of ROS-producing neutrophils, but not monocytes, in CD169-DTR kidneys when compared with the WT (Fig 5A and B). Correspondingly, these neutrophils in CD169-DTR kidneys generated significantly lower level of ROS than those in the WT, suggesting lower neutrophils' killing ability (Fig 5C and D).

We next sought to determine whether the diminished candidacidal function in CD169-DTR kidneys was contributed by

neutrophils' reduced viability. Surprisingly, both infected WT and CD169-DTR kidneys showed similar proportion of PI[neg] annexin V[+] apoptotic and PI[+] annexin V[+] dead neutrophils (Fig 5E and F). This indicates that neutrophils' viability is not one of the factors contributing to their compromised candidacidal function in CD169-DTR kidneys.

### Renal IFNγ expression was significantly lower in the absence of CD169[++] macrophages

The association of inefficient *Candida* clearance and decreased ROS levels in neutrophils of CD169-DTR kidneys prompted us to investigate the expression level of IFNγ in infected WT and CD169-DTR kidneys at day 6 p.i. This cytokine has been known to be protective against invasive candidiasis by potentiating neutrophils' candidacidal ability (Diamond et al, 1991; Kullberg et al, 1993; Stevenhagen & van Furth, 1993; Nader-Djalal & Zadeii, 1998). Intriguingly, we observed diminished renal IFNγ expression in

infected CD169-DTR kidneys by qPCR (Fig 6A), as well as by flow cytometry analysis (Fig 6B).

### IFNγ-producing cells are restricted to a CD19[int] kappa-light chain[+] cell population

Next, we attempted to delineate the type of immune cells that produce IFNγ at day 6 p.i. (Fig 7A). To our surprise, these IFNγ-producing cells are Ly6C[int]Ly6G[lo], F4/80[lo]CD11b[lo], MHCII[+]CD11c[lo], CD49b[neg]CD3[neg], and CD8[neg]CD4[neg], which indicates that they are not T lymphocytes, classical APCs and granulocytes. Instead, these IFNγ-producing cells, which are reduced distinctly in infected CD169-DTR kidneys (Fig 7B), express MHCII, kappa-light chain, and CD19[int], suggesting a B-cell–like population (Fig 7A).

Because NK cells are important modulators for potentiating phagocytes' antifungal mechanisms by IFNγ secretion in *Candida* infections (Bhatnagar et al, 2010; Costantini et al, 2010; Bar et al, 2014; Voigt et al, 2014), we last sought to reaffirm if NK cells are

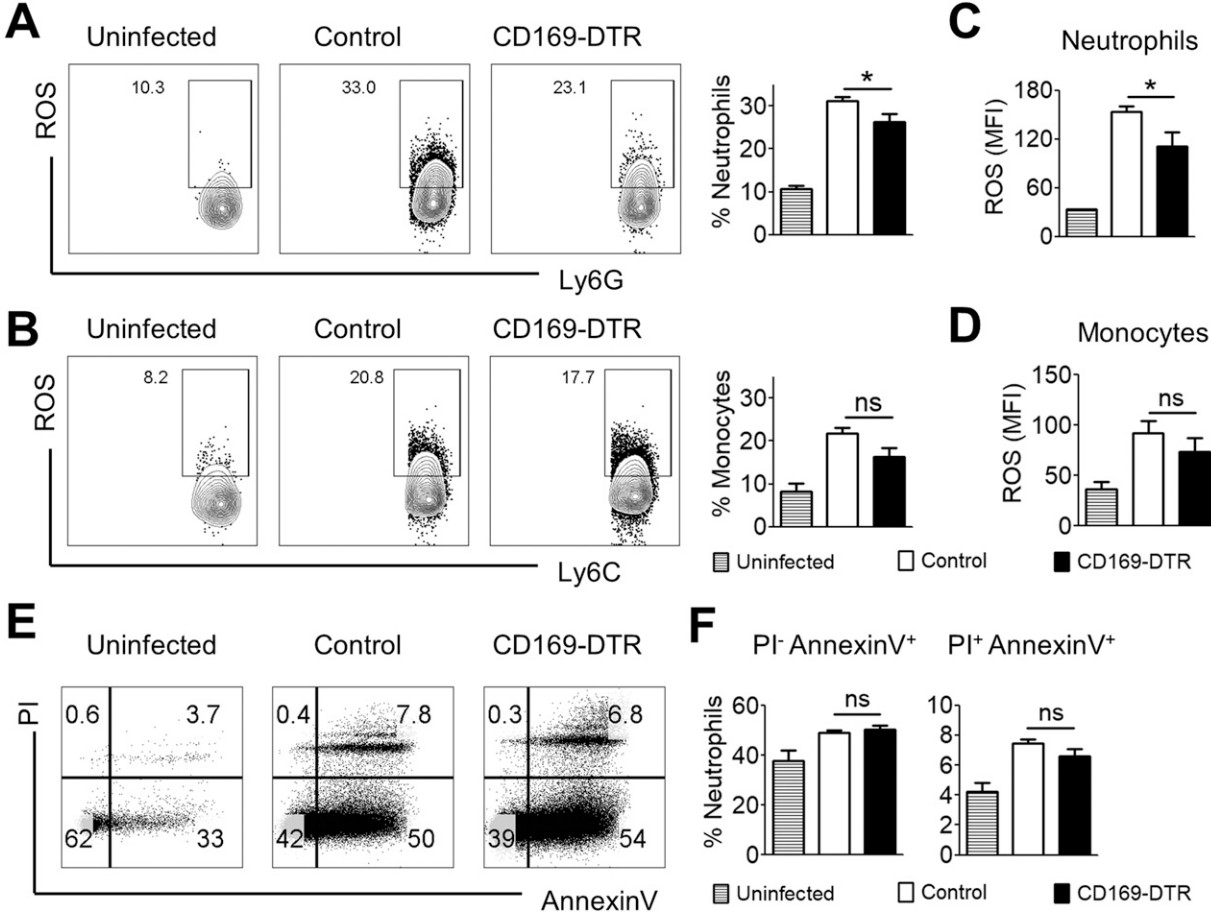

**Figure 5.  Neutrophils in CD169-DTR kidneys display lower reactive oxygen species (ROS) production.**
**(A, B, C, D)** Single-cell suspension prepared from kidneys of uninfected, infected control, and CD169-DTR mice (day 6 p.i.) were incubated with 2.5 $\mu$g/ml H$_2$DFFDA and 100 $\mu$g/ml zymosan for 60 min at 37°C. **(A, B, C, D)** Neutrophils (A, C) and monocytes (B, D) were then analyzed for fluorescein/FITC (H$_2$DFFDA converted to fluorescein by ROS) expression. **(A, B)** Representative flow–cytometry profiles of ROS-producing neutrophils (A) and monocytes (B) isolated from uninfected, infected control and CD169-DTR kidneys. **(A, B)** Bar graphs (right) represent percentage of neutrophils (A) and monocytes (B) expressing ROS. *t* test (two-tailed). **(C, D)** Bar graphs represent normalized MFI of ROS level in neutrophils (C) and monocytes (D). *t* test (two-tailed). **(E)** Representative flow cytometry profiles of propidium iodide (PI) and annexin V–stained neutrophils isolated from uninfected, infected control, and CD169-DTR kidneys (day 6 p.i.). **(F)** Bar graphs represent the frequency of PI[−]annexin V[+] (apoptotic) and PI[+] annexin V[+] (apoptotic and necrotic) neutrophils. *t* test (two-tailed).

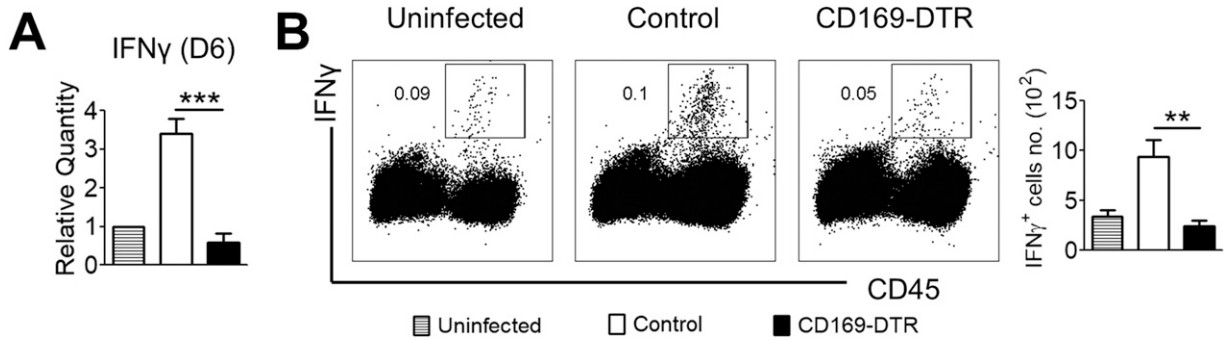

**Figure 6. Absence of CD169++ tissue-resident macrophages leads to diminished renal IFNγ response.**
**(A)** Relative transcript level of IFN-γ in the kidneys of uninfected, infected control, and CD169-DTR mice (day 6 p.i.). *t* test (two-tailed). **(B)** Representative flow cytometry profiles of IFNγ expression by CD45+ cells (left) and absolute number of IFNγ+ cells (right) in kidneys isolated from uninfected, infected control, and CD169-DTR mice (day 6 p.i.). *t* test (two-tailed). Data are shown as mean ± SEM. *P < 0.05. **P < 0.01. ***P < 0.001. ns, not significant. Data represent two (*n* = 3) independent experiments.

required to regulate the production of IFNγ in this infection. To this end, we immunodepleted NK cells in vivo and assessed the expression level of IFNγ in infected kidneys at day 2 and 6 p.i. Our data revealed that depletion of NK cells did not affect the production of IFNγ nor the amount of IFNγ–producing cells (Fig S4A–C).

## Discussion

TRMs mediate important innate responses in various infectious diseases and are generally recognized as F4/80hi and CD11b+ populations. Sialoadhesin, also known as CD169, was highlighted to be expressed only by a subset of TRM in the kidney (Karasawa et al, 2015); however, there was no subsequent report about renal CD169+ TRM. Our data support recent findings that renal TRMs are not homogenous and they can be broadly subcategorized into two subsets, namely, F4/80++

CD11b+ (CD169++) TRMs and F4/80+++ CD11b++ (CD169+) TRMs. Using CX3CR1gfp/gfp and CD169-DTR mice, we concluded that both CD169++ TRM and CD169+ TRM are indispensable for systemic *Candida* immunity, where each subset exhibits differential functions in *Candida* renal immunity. Specifically, CD169++ TRM appears to be vital in controlling renal *C. albicans* outgrowth at the later stage of the infection, whereas CD169+ TRM appears to be crucial in renal first-line defense by limiting fungal growth at the initial stage of infection. Correspondingly, lack of CD169++ TRM resulted in a distinct reduction of ROS production in neutrophils and renal IFNγ. Further investigations are warranted to understand the mechanisms in the function of renal CD169++ TRM in controlling fungal outgrowth and IFNγ production.

Our data showed that kidney is the primary target organ where *C. albicans* persists, and this is consistent with previous reports that indicate the importance of renal immunity in experimental disseminated candidiasis and subsequent mouse mortality (Spellberg

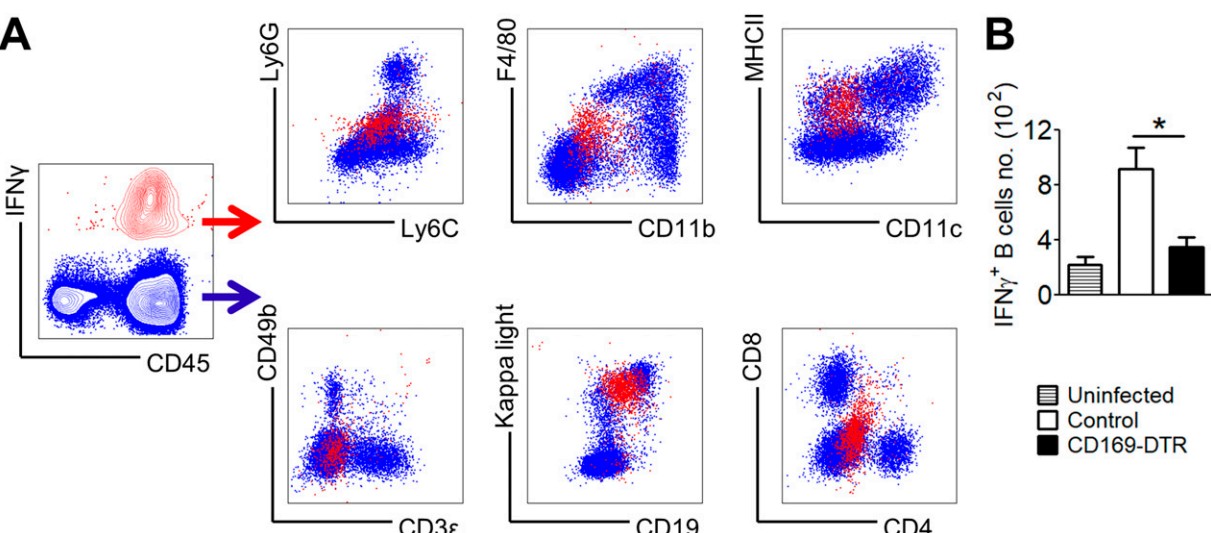

**Figure 7. *Candida*-induced renal IFNγ is not produced by classical IFNγ-producing cells.**
IFNγ-producing cells were analyzed in infected WT and CD169-DTR kidneys at day 6 p.i. **(A)** Contour plot of kidney cells stained for IFN-γ and CD45 (left). Expression profile of a panel of surface markers by IFNγ+ cells (red dots) and IFNγ− cells (blue dots). IFNγ+ cells can be identified as Ly6CintLyG lo, F4/80loCD11blo, MHCII+CD11clo, CD49bnegCD3neg, kappa-light chain+CD19int, and CD8negCD4neg. **(B)** Absolute number of IFNγ+ kappa-light chain+CD19int cells in kidneys isolated from uninfected, infected, control and CD169-DTR mice (day 6 p.i.). *t* test (two-tailed). *P < 0.05.

et al, 2003; MacCallum & Odds, 2005; Lionakis et al, 2011; Navarathna et al, 2012, 2019; Hebecker et al, 2016). In particular, despite the ablation of TRMs in all peripheral organs in CD169-DTR mice, kidneys remained as the only organ with persisting *C. albicans*, hence highlighting the distinctive role of kidney TRMs in systemic *Candida* immunity. Although kidney being the main target organ may not be consistently reflected in human counterparts, renal immunity is still believed to be an integral part of the host resistance against this infection because serious renal *Candida* infection has been observed in immunocompromised patients (Lionakis, 2014; Xu & Shinohara, 2017).

The uncontrolled *C. albicans* growth in the kidneys has previously been reasoned to be attributed by the slow or delayed recruitment of innate myeloid cells (Lionakis et al, 2011). However, in our infection model, we observed functional infiltration of immune cells in the kidneys of WT mice. In addition, CXCL1 and CCL2 in infected WT kidneys were described to be detectable as early as 12 h after infection, supporting rapid recruitment of innate myeloid cells (MacCallum et al, 2009). Recruitment of immune cells has also been reported to be regulated by macrophages, DCs, epithelial, and endothelial cells (Netea et al, 2002; Tuite et al, 2004; Tran et al, 2015). Specifically, macrophages have been shown to recruit neutrophils in the early stage of infection via CXCL2 expression (Kanayama et al, 2015; Xu & Shinohara, 2017). Nevertheless, CD169-DTR mice displayed similar cellular recruitment to the kidneys as that of the WT mice, suggesting that the orchestration of cellular infiltration is not initiated or assisted by CD169++ TRMs.

Recently, Karasawa et al (2015) reported a small subset of renal Ly6C$^{lo}$ monocytes residing at the vasculature that expresses CD169 (Karasawa et al, 2015), and together with CD169+ macrophages, elicit a critical role in preventing excessive inflammation in renal ischemia-reperfusion injury. Vascular-associated TRMs have also been described in other organs such as intestine (Honda et al, 2020) and adipose tissue (Silva et al, 2019), where they support vascular tone and integrity (unpublished data); therefore vascular-associated monocytes/TRMs could potentially have similar role in the kidney and contribute to the progression of systemic candidiasis.

Understanding of renal macrophages has been limiting due to the lack of available tools and phenotypic markers to delineate these mononuclear phagocytes (Gottschalk & Kurts, 2015; Kurts et al, 2020). Despite the attempts to understand renal macrophages through lineage-tracing and transcriptional studies (Hochheiser et al, 2013; Cao et al, 2015; Munoz et al, 2019; Liu et al, 2020; Salei et al, 2020), much less has been known for the functions of renal TRM subsets in infectious disease contexts. Although renal macrophages have previously been shown to be capable of directly limiting *C. albicans* growth and invasion into the renal tubules, our data suggest that CD169++ TRM subset does not participate in these early protective mechanisms (Marcil et al, 2002; Lionakis et al, 2013; Munoz et al, 2019). Instead, data from CX3CR1$^{gfp/gfp}$ mice suggests that CD169+ TRM subset is the main effector population for providing initial renal defense against *C. albicans* assault.

Neutrophils are the integral innate effector cells in *Candida* immunity (Fulurija et al, 1996; Aratani et al, 1999; Netea et al, 2015) and patients suffering from neutropenia and neutrophil defects are at higher risks for invasive candidiasis (Lionakis & Netea, 2013). The importance of oxidative killing effector pathway is underlined by the findings that chronic granulomatous disease and myeloperoxidase

deficiency, where ROS production is defective in neutrophils, impair *C. albicans* killing in both patients and mice (Aratani et al, 1999, 2002, Lehrer & Cline, 1969). Correspondingly, absence of CD169++ TRMs that led to a significant reduction of ROS+ neutrophils, and surprisingly not monocytes, suggests that the uncontrolled growth of renal *C. albicans* was likely due to reduced neutrophils' fungal killing capacity.

The importance of IFNγ in systemic candidiasis has been well depicted in experimental murine model, wherein IFNγ deficiency in knockout mice increases their susceptibility to this infection (Balish et al, 1998; Cenci et al, 1998; Kaposzta et al, 1998). Furthermore, it has been demonstrated that administration of IFNγ increases neutrophils' and macrophages' phagocytic and fungal killing capacity, which in turn improves patients' resistance against invasive candidiasis (Djeu et al, 1986; Malmvall & Follin, 1993; Marodi et al, 1993). In our TRM depleting mouse model, we consistently observed downregulation of renal IFNγ in the absence of CD169++ TRM, which suggests the involvement of CD169++ TRM in either the initiation or maintenance of IFNγ expression. While the protective role of IFNγ in disseminated candidiasis is evident, IL17-mediated protective response in this model has mostly been controversial (Balish et al, 1998; Lavigne et al, 1998; MacCallum, 2009; Kashem et al, 2015; Mengesha & Conti, 2017). Nevertheless, IL17-mediated immunity has been known to be essential for oral and dermal candidiasis (Conti & Gaffen, 2010; Netea et al, 2015; Mengesha & Conti, 2017). In our model, the ablation of CD169++ TRM did not affect IL17 expression level. In fact, we did not observe much IL17 expression in infected WT kidneys (data not shown). Similar to our observations, LeibundGut-Landmann et al (2007) reported significantly higher amount of IFNγ than IL17 in infected WT mice (LeibundGut-Landmann et al, 2007), indicating that IFNγ-mediated immunity is more pronounced in systemic candidiasis. Another study that corroborated the beneficial role of IFNγ in *Candida* immunity was the protection elicited by the adoptive transfer of IFNγ-producing Th1 cells, but not IL17-producing Th17 cells, against systemic candidiasis (Kashem et al, 2015).

As a first step toward unraveling the intriguing correlation among CD169++ TRM, IFNγ and neutrophils, our pilot experiment sought to reveal the cell source of IFNγ in our model. NK cells, being early IFNγ – producers, have been shown to be capable of recognizing *C. albicans* and potentiating neutrophils' fungal killings through the secretion of IFNγ and GM-CSF (Bhatnagar et al, 2010; Bar et al, 2014; Voigt et al, 2014; Whitney et al, 2014; Domínguez-Andrés et al, 2017). In addition, it has been shown that 5–10% of the NK cells produce IFNγ in *Candida*-infected kidneys (Whitney et al, 2014). To our surprise, we did not observe IFNγ+ NK cells nor did we observe down-regulation of IFNγ in the absence of NK cells in our model. This observed discrepancy could be due to the assessment of IFNγ-producing cells at different time points. Specifically, we assessed the IFNγ production at day 6 p.i., a time point that was much later as compared with Whitney's group, that is, 16 h p.i. Hence, NK cells may not be involved in the maintenance of IFNγ production for this infection. Interestingly, Murciano et al reported that killed *C. albicans* yeasts inhibit IFNγ release by NK cells (Murciano et al, 2006). Another probable reason for the discrepancy could be the experimental method used for investigating IFNγ secretion. In this article, monesin, a protein secretion inhibitor, was administrated directly to the mice and the cellular IFNγ production was subsequently assessed without ex vivo cell incubation. On the other hand, in the study by Whitney et al (2014), before IFNγ

analyses, renal cells were first prepared and incubated in culture with a protein secretion inhibitor for 7 h (Whitney et al, 2014). Surprisingly, other lymphoid and myeloid subsets investigated in our study, in particular CD169++ TRM, also do not appear to express IFNγ, despite studies reported direct secretion of IFNγ by macrophages (Bogdan & Schleicher, 2006). Instead, our data suggest that the major IFNγ producers are restricted to a B-like cell population. Intriguingly, a similar subpopulation of innate B cells producing IFNγ has been reported to be vital in regulating and initiating early innate immune response of intracellular bacterial infections (Bao et al, 2014; Krocova et al, 2020). In addition, other prior works revealed that mature B cells, when primed by T cells and stimulated by pathogens or TLR ligands, can secrete IFNγ (Gray et al, 2007; Lund & Randall, 2010). Although it is not uncommon for B cell–IFNγ axis in microbial infections, future studies are urgently needed to comprehensively profile this renal IFNγ+ B-cell–like population and identify its relationship with CD169++ TRM in the kidney.

In summary, we conclude that renal TRMs can be sub-classified into two major populations based on the CD169 expression. More importantly, these two subsets harbor non-redundant protective functions in disseminated candidiasis, which provide insights into the cellular basis of protective host innate immunity against *C. albicans*. These revelations should be useful in the future design of therapeutic interventions.

# Materials and Methods

### Mice

CD169-DTR transgenic mice were generated in our laboratory in BALB/c genetic background as described previously (Purnama et al, 2014), and subsequently cross bred with C57BL/6 for 10 generations. CX3CR1gfp/gfp mice were purchased from The Jackson Laboratory. CD169-DTR C57BL/6 transgenic mice, together with WT C57BL/6, were bred and maintained under specific pathogen-free conditions in the Nanyang Technological University (NTU) animal facility. Male mice (7–10 wk of age) were used for *Candida* infection. All experiments were approved by the Institutional Animal Care and Use Committee under the number ARF-SBS/NIE A-0380AZ.

### *C. albicans* infection

The clinical isolate *C. albicans* strain SC5314 was cultured on yeast extract–peptone–dextrose (YPD) plate for 24 h at 30°C, followed by 16–18 h in YPD media. *C. albicans* suspension was centrifuged, washed, and counted. For infection, $5 \times 10^4$ cfu *C. albicans* were i.v. injected into each mouse. For rIFNγ treatment, infected mice were treated with 10,000 U recombinant IFNγ (R&D Systems) daily. Mice were monitored and weighed daily during the course of infection.

### Diphtheria toxin-mediated and antibody-mediated ablation

DT (10 ng/g body weight) was prepared in PBS supplemented with 1% mouse serum. CD169-DTR and WT mice were administered i.p. with DT according to the infection scheme (Fig S1).

For depletion of NK cells, mice were administered i.p. with 100 μg antimouse NK1.1 (PK136; Invivomab) antibody daily.

### Fungal burden analyses

Brain, heart, lungs, liver, kidneys, and spleen were harvested at indicated time points of infection, weighed, and minced before incubating with 1 mg/ml collagenase D in digestion medium for 1 h at 37°C. Samples were resuspended repeatedly until no visible aggregates. Serial dilution was conducted and plated on YPD plates. The plates were incubated for 48 h at 30°C. CFU was determined by manual counting of the colonies. Fungal burden was expressed as CFU per gram of organ.

### Tissue collection, processing, and isolation of single-cell suspension

Kidneys were harvested, minced and incubated with 1 mg/ml collagenase D in digestion medium for 1 h at 37°C. The minced kidneys were pipetted up and down repeatedly until there were no visible aggregates. After centrifugation at 330*g* for 5 min, the cell pellets were resuspended with 5 ml 35% Percoll (GE Healthcare Life Science) and centrifuged at 600*g* for 15 min. Supernatants were discarded and cell pellets were resuspended with 5-ml RBC lysis buffers. After lysis, cell suspensions were centrifuged and resuspended with IMDM 2%. The single-cell suspensions were stored at 4°C until further use. For cell counting, small aliquots of cell suspensions, premixed with tryphan blue, were counted using hemocytometer.

For cell sorting, kidney single-cell suspensions were incubated with rat anti-mouse CD16/32 (2.4G2, 1 mg/ml), followed by fluorescent antibodies against surface antigens mouse CD45 (30F11), F4/80 (BM8), and CD11b (M1/70). Stained cells were washed once and passed through a 40-μm filter before sorted on a 4-laser BD FACSAria II cell-sorter (BD Bioscience).

### Flow cytometry analyses

Single-cell suspensions were incubated with rat anti-mouse CD16/32(2.4G2, 1 mg/ml) at 4°C for 15 min (1:100) in FACS buffer (PBS supplemented with 2% FCS) to block Fc receptors. For staining of surface antigens, the cells were incubated with fluorochrome-conjugated antibodies against mouse CD45 (30F11), F4/80 (BM8), CD11b (M1/70), Ly6G (1A8), and Ly6C (HK1.4) at 4°C for 20 min. Stained cells were washed once and resuspended in FACS buffer for acquisition on a five-laser BD LSRFortesssa X-20 (BD Biosciences). For detection of intracellular cytokine IFNγ, kidney single-cell suspensions were prepared from mice treated with protein secretion blocker monensin according to the protocol (Sun et al, 2009). Briefly, each mouse was injected i.v. with 250 μl of PBS containing 100 μg of monensin (M5273; Sigma-Aldrich), 5 h before organ isolation and subsequent preparation of kidney single-cell suspensions. The cells were incubated with rat anti-mouse CD16/32 antibody followed by staining with fluorescent antibodies against surface antigen mouse CD45 (30F11), CD11b (M1/70), F4/80 (BM8), Ly6G (1A8), Ly6C (HK1.4), CD3ε (145-2C11), CD49b (HMa2), CD19 (1D3), CD4 (GK1.5), CD8 (53.6–7), CD11c (N418), MHCII (M5/114.15.2), and Ig κ

light chain (RMK-45). Stained cells were washed once with FACS buffer followed by fixation (2% PFA) and permeabilization (0.05% saponin) before incubating with anti-IFNγ antibody (1:500; BioLegend) in 0.05% saponin solution. Stained cells were washed once and resuspended in FACS buffer for acquisition on a five-laser BD LSRFortesssa X-20 (BD Bioscience). For detection of cell death and apoptosis, kidney single-cell suspensions were initially incubated with rat anti-mouse CD16/32 antibody followed by staining with fluorescent antibodies against surface antigen mouse CD45 (30F11), CD49b (HMa2), CD3ε (145-2C11), CD11b (M1/70), Ly6G (1A8), and Ly6C (HK1.4). Stained cells were washed once with FACS buffer and 1x annexin V binding buffer (BioLegend) before incubating with FITC-conjugated annexin V according to the manufacturer's instructions (BioLegend). Stained cells were washed once and resuspended in 1× annexin V buffer containing propidium iodide (PI, 1 $\mu g$/ml) before acquisition on a five-laser BD LSRFortessa X-20 (BD Bioscience).

### ROS detection assay

Single-cell suspensions were washed and incubated with 2.5 $\mu g$/ml $H_2$DFFDA, with or without 100 $\mu g$/ml zymosan, for 60 min at 4°C or 37°C. The cells were then washed two times with IMDM 2% and stained with fluorochrome-labeled antibodies at 4°C for 20 min. Stained cells were washed and resuspended in FACS buffer containing PI for acquisition on a five-laser BD LSRFortessa X-20 (BD Bioscience). Data were normalized based on the corresponding cells incubated in 4°C.

### Serum creatinine

Blood sera were collected from mice at day 10 p.i. Serum creatinine levels were quantified using the mouse Cr (Creatinine) ELISA Kit (Elabscience), according to the manufacturer's instructions.

### Evans blue assay

Blood vessel permeability was assessed in accordance to Radu and Chernoff (2013). Briefly, uninfected and infected WT and CD169-DTR mice were injected i.v. with 200 $\mu l$ of 0.5% Evans blue/PBS. 30 min later, the mice were euthanized and their organs were collected, weighed, and incubated in 100% formamide in microfuge tubes for 48 h. Evans blue-infused formamide (0.5 ml) were transferred to disposable polystyrene cuvettes without transferring any tissue pieces. Absorbances at 610 nm were recorded, and these values were normalized by the weight of the organs.

### Histological analyses

For immunofluorescence staining, harvested kidneys were embedded in Optimal Cutting Temperature compound (O.C.T. Tissue Tek) and stored at −80°C. 6-μm sections were cut and fixed in acetone for 10–15 min. Sections were incubated with Fc-block for 20 min, washed, and incubated with fluorescent antibodies for 1 h. Washed sections were then mounted with DAKO fluorescent mounting medium. Images were obtained using a Nikon Eclipse 80i microscope at 20× objective magnification.

For H&E and PAS stainings, culled mice were perfused with 4% PFA (Sigma-Aldrich). Harvested kidneys were incubated in 4% PFA for 24 h at RT, dehydrated, and embedded in paraffin (Paraplast

Plus; Leica). Paraffin-embedded blocks were then sectioned into 6-μm sections. Sections were deparaffinized in xylene for 20 min and rehydrated. For hematoxylin and eosin (H&E) staining, the sections were stained with modified Mayer's solution hematoxylin (Abcam) and counterstained with eosin (Sigma-Aldrich). After clearing with xylene, the sections were mounted with DPX Mountant (Sigma-Aldrich). For PAS staining, the sections were incubated with periodic acid solution (Abcam), washed, and stained with Schiff's solution (Abcam). After this, the sections were counterstained with modified Mayer's solution hematoxylin (Abcam). After clearing with xylene, the sections were mounted with DPX mountant (Sigma-Aldrich). The images were obtained using Eclipse 80i microscope (Nikon) with the Digital Sight DS-U3 (Nikon) and NIS-Elements D software (Nikon) at 4× 10×, 20×, and 100× magnifications.

### Quantitative real-time PCR

Harvested kidneys were immediately homogenized in TRIzol (Thermo Fisher Scientific) using a homogenizer (CAT X360). RNAs were extracted according to the manufacturer's instructions (RNAsimple Total RNA kit; Tiangen Biotech Ltd). Single-strand cDNA synthesis from the purified RNA was performed in accordance to the manufacturer's instructions (Promega M-MLV reverse transcriptase). Real-time PCR was performed according to the manufacturer's instructions using the Primerdesign Precision FAST protocol (Primerdesign Ltd).

Primer sequences were as follows: *CXCL1*; Fwd: ACTGCACCCAAACCGAAGTC, Rev: TGGGGACACCTTTTAGCATCTT. *CXCL2*; Fwd: ACAGAAGTCATAGCCACTCTC, Rev: CCTTGCCTTTGTTCAGTATC. *CCL2*; Fwd: CATCCACGTGTTGGCTCA, Rev: GATCATCTTGCTGGTGAATGAGT. *TNF-α*; Fwd: TCTTCTCATTCCTGCTTGTGG, Rev: GGTCTGGGCCATAGAACTGA. *G-CSF*; Fwd: GTGCTGCTGGAGCAGTTGT, Rev: TCGGGATCCCCAGAGAGT. *GM-CSF*; Fwd: GCATGTAGAGGCCATCAAAGA, Rev: CGGGTCTGCACACATGTTA. *M-CSF*; Fwd: GGTGGAACTGCCAGTATAGAAAG, Rev: TCCCATATGTCTCCTTCCATAAA. *IL10*; Fwd: CAGAGCCACATGCTCCTAGA, Rev: TGTCCAGCTGGTCCTTTGTT. *ICAM-1*; Fwd: AGTCCGCTGTGCTTTGAG, Rev: AGGTCTCAGCTCCACACT. *VCAM-1*; Fwd: TCTTACCTGTGCGCTGTGAC, Rev: ACTGGATCTTCAGGGAATGAGT. *KIM-1*; Fwd: AGATACCTGGAGTAATCACACTGAAG, Rev: TGATAGCCACGGTGCTCA. *CD169*; Fwd: GCTGATACTGGCTTCTACTTCT, Rev: AGGTGGTCAGGTCTGGAGTAA. *IFN- γ*; Fwd: CACGGCACAGTCATTGAAAG, Rev: CCAGTTCCTCCAGATATCCAAG. *β-actin*; Fwd: AAGGCCAACCGTGAAAAGAT, Rev: CTGTGGTACGACCAGAGGCATACA. *hHB-EGF*; Fwd: ATGACCACACAACCATCCTG, Rev: CCAGCAGACAGACAG ATGACA.

### Computer software

FACS analyses were conducted using FlowJo X 10.0.7 software (Tree Star, Inc.). Graphs and statistical analyses were generated using GraphPad Prism 5.0 software (GraphPad Software). Image processing was performed using ImageJ 1.48 software (Wayne Rasband; National Institutes of Health).

### Statistical analyses

Statistical significance was tested by unpaired two-tailed *t* test, one-way ANOVA with Bonferroni's multiple comparisons tests and

two-way ANOVA with Bonferroni posttests as stated accordingly in the figure legends. Survival curves were analyzed by the Mantel–Cox long-rank test. Statistical significance is demonstrated in the figures with asterisks: $*P < 0.05$, $**P < 0.01$, $***P < 0.001$.

## Data Availability

The original flow cytometry data have been deposited in the NTU Open Access Data Repository (Digital Repository-NTU).

## Supplementary Information

## Acknowledgements

The authors would like to thank Monika Tetlak for the mouse management and Prof. Wang Yue (IMCB, A*star) for the *Candida* strain (SC5314). This work was supported by the Ministry of Education Tier 2 grant (MOE2016-T2-1-012) awarded to C Ruedl.

### Author Contributions

YJ Teo: conceptualization, data curation, formal analysis, investigation, methodology, and writing—original draft, review, and editing.
SL Ng: methodology and writing—review and editing.
KW Mak: methodology.
YA Setiagani: methodology.
Q Chen: methodology.
SK Nair: methodology.
J Sheng: formal analysis in review.
C Ruedl: conceptualization, supervision, funding acquisition, project administration, and writing—review and editing.

### Conflict of Interest Statement

The authors declare that they have no conflict of interest.

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
