## [Reviewer comments · Life Science Alliance]

Life Science Alliance

Renal CD169⁺⁺ resident macrophages are crucial for protection against acute systemic candidiasis

Christiane Ruedl, Yi Juan Teo, See Liang Ng, Keng Wai Mak, Yolanda Aphrilia Setiagani, Qi Chen, Sajith Kumar Nair, and Jianpeng Sheng

DOI: <https://doi.org/10.26508/lsa.202000890>

Corresponding author(s): Christiane Ruedl, Nanyang Technological University and Yi Juan Teo, Nanyang Technological University

Review Timeline:

Submission Date:	2020-08-21
Editorial Decision:	2020-09-25
Revision Received:	2020-11-25
Editorial Decision:	2020-12-23
Revision Received:	2021-01-18
Editorial Decision:	2021-01-19
Revision Received:	2021-01-22
Accepted:	2021-01-22

Scientific Editor: Shachi Bhatt

Transaction Report:

September 25, 2020

Re: Life Science Alliance manuscript #LSA-2020-00890-T

Prof. Christiane Ruedl
Nanyang Technological University
School of Biological Sciences
60 Nanyang Drive
Singapore 637551
Singapore

Dear Dr. Ruedl,

Thank you for submitting your manuscript entitled "Renal CD169++ tissue-resident macrophages are crucial for protection against acute systemic candidiasis" to Life Science Alliance (LSA). The manuscript has been reviewed by the editors and outside referees (reviewer comments below). As you will see, the reviewers were enthusiastic about the study and its potential impact, but have raised a number of concerns that should be addressed prior to further consideration of the manuscript at LSA. In particular, the reviewers were concerned about the results sections pertaining to the mechanism of CD169++ TRMs - effect on ROS production in PMNs and the link to IFN γ . Therefore, although we are unable to publish the current version of the manuscript, we would encourage you to submit a revised version that addresses the referees concerns. We understand that the request from Rev 3 about further data supporting the identification of a B cell like population as a source for IFN γ might be difficult to obtain - we encourage you to address this to the best of your ability, and discuss the caveats in the manuscript text.

We would be happy to discuss the individual revision points further with you, should this be helpful. A revised manuscript may be re-reviewed, most likely by some or all of the original referees. When submitting the revision, please include a letter addressing the reviewers' comments point-by-point and a copy of the text with alterations highlighted (boldfaced or underlined). The typical time frame for revisions is three months. In an effort to expedite the review process, papers are generally considered through only one revision cycle.

Thank you for considering Life Science Alliance as an appropriate venue for your research. We look forward to receiving your revised manuscript.

Sincerely,

Shachi Bhatt, Ph.D.
Executive Editor
Life Science Alliance

- A letter addressing the reviewers' comments point by point.
- An editable version of the final text (.DOC or .DOCX) is needed for copyediting (no PDFs).
- High-resolution figure, supplementary figure and video files uploaded as individual files: See our detailed guidelines for preparing your production-ready images, <https://www.life-science-alliance.org/authors>
- Summary blurb (enter in submission system): A short text summarizing in a single sentence the study (max. 200 characters including spaces). This text is used in conjunction with the titles of papers, hence should be informative and complementary to the title and running title. It should describe the context and significance of the findings for a general readership; it should be written in the present tense and refer to the work in the third person. Author names should not be mentioned.

B. MANUSCRIPT ORGANIZATION AND FORMATTING:

Reviewer #1 (Comments to the Authors (Required)):

In this manuscript, Teo and colleagues test the idea that different subsets of kidney tissue resident macrophages are important for preventing *C. albicans* mediated fungal infections in the kidney. They show convincing data that fraction 1 kidney resident macrophages are required for clearing the fungal infection at later time points and that fraction 2 kidney resident macrophages are required for limiting the initial fungal burden. These data are strong, novel, and convincing. However, the latter portion of this manuscript, which attempts to address the mechanism, falls a bit short. This reviewer feels that several of the pieces of data to understand mechanism are there, but the order of presentation and rationale should be altered to improve readability/quality of the manuscript. This manuscript could also be strengthened by a thorough proof-reading as well as a one or two additional experiments. The most critical of these additional experiments is a simple characterization of CD169+ and CD169++ cells (and analysis of CD169 expression in individual cells). This may also increase the authors understanding of how different fractions of tissue resident macrophages restrict fungal burden through different mechanisms. A list of detailed

suggestions and comments is listed below.

Specific comments:

It is difficult to tell what region of the kidney is shown in Figure 1C. The authors should either include markers that identify cells from either region or show a view of the kidney where cortex and medulla can be clearly identified.

If all TRM express CD169 to some level as shown in figure 1D, would the authors not expect a greater reduction in both fraction 1 and fraction 2 following DT treatment? Do fraction 2 cells express the DTR on the cell surface? Is it possible that a few cells in fraction 2 express high CD169 while a majority express none? This seems to make more sense than the idea that all cells in fraction 2 express intermediate levels of Cd169.

This paper would be significantly strengthened if a more thorough characterization (multi-parameter flow; RNA seq) of CD169⁺⁺ and CD169⁺ resident macrophages was performed. This includes a better analysis of how much CD169 is expressed in individual fraction 1 and fraction 2 resident macrophages through multi-parameter flow cytometry.

If the fungal burden is the same in WT and CD169 DTR mice at day 6, why is there differences in kidney damage (tubular necrosis, hemorrhages, etc)? Are all of the *C albicans* alive in both groups? This should be addressed.

The data presented in figure 5I claiming that "Exogenous rIFN γ rescued neutrophils' oxidative killing in infected CD169-DTR mice" is a bit of a stretch as Ifng treatment only partially rescues ROS production. Does treatment with Ifng change the % of neutrophils that produce ROS? This would help to convince me that Ifng does indeed stimulate ROS production in neutrophils.

Why the immediate focus on NK cells in figure 6? Several cells can produce Ifng including Th1 cells, ILC1s, NKT1s, Nk cells, and macrophages. It is unclear as to why the authors jump to exclude NK cells as being important without first identifying the major cell type that produces Ifng, which they show are kappa⁺ B cells in the last figure. This figure is confusing and should be removed (or at the very least, moved to supplementary).

In my opinion, the authors should make figure 5 G-J a separate figure. Figure 5 will show that ROS production is defective in CD169 deficient mice. New figure 6 would show that an inducer of ROS, Ifng, is also defective and that treatment with Ifng partially rescues ROS production. It is my opinion that the authors should then show the major cell type that is producing Ifng, which are kappa⁺ B cells. The major experiments that are needed to decisively show the mechanism are: 1) to show that B cell specific Ifng production is lost in CD169⁺⁺ DTR mice and 2) to show that loss of kappa⁺ B cells (or Ifng produced by these cells) mimics the phenotype found in CD169⁺⁺ deficient mice. However, this may be beyond the scope of this manuscript.

The section titled "Fraction I and Fraction II populations displayed differential expression profiles." comes out of nowhere and doesn't fit with the story. Why do the authors show expression of CX3CR1, Il12p35/p40 and Il10 here? If the authors are trying to characterize fraction 1 and fraction 2 macrophages from a wild type mouse, as they should, they should include this data earlier. RNA sequencing or multiparameter flow cytometry of fraction 1 and fraction 2 resident macrophages would be much more informative. This goes with the comment from above.

Can CD169⁺ macrophages directly influence ROS production by neutrophils and is this dependent

on Ifng? This is especially relevant since Ifng treatment only mildly rescued ROS production in neutrophils in figure 6A.

Reviewer #2 (Comments to the Authors (Required)):

The authors provide convincing evidence that CD169+ macrophages are required for the defense against renal candidiasis in the mouse. This is an interesting finding worth reporting, since little is known about these macrophages in the kidney, on their role in the defense against kidney infection and on a role against candidiasis.

There are however a number of questions that need to be addressed before acceptance can be recommended. Most notably, the mechanistic clarification lacks depth.

Major concerns

1) the authors should characterize their depletion model more carefully. They only showed how many macrophages defined by different F4/80 and CD11b expression levels were depleted. First, they should show how efficiently CD169+ cells were depleted in the kidney and other organs. Next, it is essential to show selectivity of depletion, because if other cell types are depleted as well, conclusions on macrophages are not justified any more. This should be done under homeostatic conditions and infection, where CD169 expression might differ, and other cell types might become susceptible to depletion.

2) Previous studies by others have located CD169+ cells in the vicinity of real vessels. Fig 1C is intended to show their location in the present study. However, this immunofluorescence microscopy shows no topological information. The authors should use immunohistochemistry.

3) the authors only used wild type mice injected with diphtheria toxin as controls. The control of PBS injected 169-DTR mice is lacking. This does not need to be repeated for all experiments, it is sufficient to show in one experiment that this transgenic mouse line is not more susceptible to candida than wild type mice

4) it remains unclear how CD169+ macrophages contribute to the defense against candida. The authors speculate that an observed reduction in IFNg is responsible, and identify not further characterised immune cells as potential producers. First, they did not provide evidence that the lack of IFNg is causal. The authors show that other inflammatory cytokines are reduced as well (Fig S3). Why then IFNg, and not these other cytokines?

And even if IFNg was causal, the connection to CD169+ macrophages is unclear. They did not show that the depletion of CD169+ macrophages reduces their IFNg production. They also did not identify the molecular mechanism by which macrophages induce IFN production in these unidentified cells, but that is probably beyond the scope of the present study.

Minor concerns

- it should be mentioned that human kidneys are not as susceptible to candidiasis as mouse kidneys. Serious kidney candida infection is only seen in immunocompromised patients
- How much were Cd169+ cells reduced in CX3CR1 Kos?
- Fig 4: Monocytes were defined as Ly6C+ CD11b+ cells but that includes also neutrophils

Reviewer #3 (Comments to the Authors (Required)):

In this manuscript by Ruedl and colleagues, the authors have investigated the roles of CD169-expressing cells in invasive candidiasis using a CD169-DTR mouse. The authors describe that the mice lacking the CD169-expressing cell-types are susceptible to invasive candidiasis and display enhanced kidney damage and inflammation. Based on the data, the authors surmise that the CD169-expressing cells regulate IFN-g production by B-cell like cells and in turn regulate the PMNs' candidacidal activity. Although the role of CD169-expressing cells during invasive candidiasis has not been characterized before and therefore the work of potential interest, there are several concerns regarding the data and the authors' conclusions that are not fully supported by the findings. I have outlined my major concerns below:

1. CD169 has been shown to be expressed on a subset of renal tissue resident macrophage, hence the rationale for using a CD169-DTR mouse to assess this macrophage population is meritorious. In addition to the above resident macrophages, monocytes have also been shown to express CD169 (PMID: 25266072). Hence, with the authors' approach, DT administration should ablate these monocytes as well. However, based on their data this does not seem to be the case. The authors should address this discrepancy. Additionally, it would be ideal if the authors provide a portrait of different cell-types which express CD169 during invasive candidiasis using FACS. This will help the readers understand the target cells of the DT depletion of CD169 in their model.

2. The authors surmise that CD169⁺⁺ macrophages regulate PMNs' candidacidal activity. This conclusion is solely based on their observation that the PMNs from CD169-DTR mice have only slightly lower ROS production capacity. Hence, impaired candidacidal activity is indirectly inferred from the decreased ROS production capacity. The direct *Candida* killing has not been examined at all. Several approaches to examine *Candida* killing has been described (PMID: 26791948; 32554707; other papers). The authors can examine PMNs-mediated *Candida* killing to directly support their claims. As of now, the claim that PMN candidacidal activity is impaired is not supported by the data.

3. In vitro studies have pointed towards protective roles of IFN-g, however IFN-g in invasive candidiasis has not been shown to be protective in conventional non-gnotobiotic animals (PMID: 9125557). In addition, patients with IFN-g signaling defect do not develop invasive candidiasis, they are at risk for mycobacterial and endemic fungal infections. Hence, the authors' rationale of assessing IFN-g deficiency as sole driver of susceptibility in CD169-DTR needs clarification and importantly, a decrease in IFN-g in CD169-depleter mice is unlikely to explain the dramatic increase in mortality in the model. Do IFN-g KO mice in the authors' hands phenocopy CD169-depleter mice? Can they rescue the phenotype of the CD169-depleter mice with recombinant IFN-g? Other mechanisms beyond IFN-g are likely operational to explain the susceptibility of CD169-depleter mice in the model.

4. In the final figures, the authors analyze IFN-g cellular sources and conclude that a B cell like population is responsible for IFN-g production. In order to support their conclusion, the authors need substantial amount of evidence and work. Some suggestions as below:

a. Independent assessment of IFN-g production capacity by this specific cell-type in context of invasive candidiasis to rule out technical artifact (autofluorescence; fluorochrome overlap), which seems more likely than a true finding that a B cell like cell is the predominant IFN-g cell in the *Candida*-infected kidney. In fact, there are not many B cells in the *Candida*-infected kidneys further raising concern that the cell that the authors claim is a B cell is likely to be a different cell that flow cytometrically appears as if it expresses B cell markers due to artifact.

b. In order to assess CD169⁺⁺ cell dependence of IFN-g production by this specific cell population,

the authors need to show whether in the CD169-DTR mice, this cell population has impaired IFN-g production.

Additionally, there are some other minor issues which also require further attention. I have highlighted them below:

1. The representative flow cytometry plot for CD169-DTR mice (and also control mice) have been used at multiple locations throughout the manuscript, this should not be the case. A same flow cytometry plot cannot represent independent experiments.
2. From Figure 1, the authors mention that the CD169⁺⁺ macrophages are present in the medullary region and are ablated in the CD169-DTR mice. However, based on their high magnification image, this is not clear. The authors should provide a larger view of the kidney where cortical/medullary/pelvic regions are clearly visible and thus the localization of the corresponding cell ablation can be visualized.
3. In Figure 1A, the FACS ancestry of the gated F4/80-CD11b cells is not clear. It should be noted on which population these cells are gated on. It would also help if "gating strategy" for their flow cytometry experiments is provided in supplement.
4. In Figure 1D and in supplements, the Fraction I and Fraction II were sorted and utilized in experiments. In the supplement, the "post-sort" plots should be provided. This would be especially essential, if the cells are being sorted based on the gates' geometry as depicted in Figure 1A, since the 2 gates are pretty close to each other. The gates' proximity can cause cross-contamination in each fraction. Without the availability of "sorting strategy" and "post-sort" it is difficult to assess the purity of each macrophage fraction.

Point-by-Point Reply

We thank all the reviewers for their constructive comments on our work. All critiques have been very helpful in preparing the revised manuscript. We have tried to address as well as we can on the experimental concerns and provide detailed answers to the raised remarks. Amendments in the manuscript have been highlighted in blue.

We hope that with the described changes and responses, the paper is now suitable for publication in LSA.

Reviewer #1:

In this manuscript, Teo and colleagues test the idea that different subsets of kidney tissue resident macrophages are important for preventing *C. albicans* mediated fungal infections in the kidney. They show convincing data that fraction 1 kidney resident macrophages are required for clearing the fungal infection at later time points and that fraction 2 kidney resident macrophages are required for limiting the initial fungal burden. These data are strong, novel, and convincing. However, the latter portion of this manuscript, which attempts to address the mechanism, falls a bit short. This reviewer feels that several of the pieces of data to understand mechanism are there, but the order of presentation and rationale should be altered to improve readability/quality of the manuscript.

This manuscript could also be strengthened by a thorough proof-reading as well as a one or two additional experiments. The most critical of these additional experiments is a simple characterization of CD169⁺ and CD169⁺⁺ cells (and analysis of CD169 expression in individual cells). This may also increase the authors understanding of how different fractions of tissue resident macrophages restrict fungal burden through different mechanisms. A list of detailed suggestions and comments is listed below.

Specific comments:

1) It is difficult to tell what region of the kidney is shown in Figure 1C. The authors should either include markers that identify cells from either region or show a view of the kidney where cortex and medulla can be clearly identified.

One of the landmark architectures of renal cortex comprise of the glomeruli, which can be detected by CD31 positive staining. CD31 is constitutively present on the endothelial linings. In the image showing the cortical region, a cluster of CD31⁺ cells indicates presence of blood vessels and hence the glomerulus. The location of CD169⁺ macrophages has previously been reported by Karasawa et al., as mentioned in the Results session - *CD169⁺⁺ macrophages are a subpopulation of renal tissue-resident macrophages* (1st paragraph, 4th row). We have included this data in Figure 1

2) If all TRM express CD169 to some level as shown in figure 1D, would the authors not expect a greater reduction in both fraction 1 and fraction 2 following DT treatment? Do fraction 2 cells express the DTR on the cell surface? Is it possible that a few cells in fraction 2 express high CD169 while a majority express none? This seems to make more sense than the idea that all cells in fraction 2 express intermediate levels of Cd169.

Yes, the DT-DTR system is sensitive and it depletes any cells that express the human heparin-binding EGF-like growth factor (hHB-EGF), the ligand of DT, upon DT administration. Thus, it is likely that the remaining Fraction 2 cells after DT treatment do not express the huDTR, hence rendering them resistant to DT. Correspondingly, Fr1 TRMs express relatively higher hHB-EGF level when compared to Fr2 TRM, hence indicating Fr1 to be more susceptible to DT treatment. This data is now added in Fig. 1E.

3) This paper would be significantly strengthened if a more thorough characterization (multi-parameter flow; RNA seq) of CD169⁺⁺ and CD169⁺ resident macrophages was performed. This includes a better analysis of how much CD169 is expressed in individual fraction 1 and fraction 2 resident macrophages through multi-parameter flow cytometry.

To confirm the presence of two main F4/80^{hi} TRM subpopulations (Fraction I and II), we have analysed scRNAseq data that was recently published by Zimmerman et al. (Zimmerman et al., 2019). Among other identified CD45⁺ immune cells (NK-cells, T cells, ILC, B cells and neutrophils), the UMAP analysis visualizes a large cell fraction of F4/80⁺ macrophages which can be subdivided in two main (0 and 1) and one minor (3) subset. Of note, a weak Siglec-1 (CD169) signal is only detectable in Fraction 0 which indicates that only one of these F4/80⁺ subsets express detectable levels of this I-type lectin, hence supporting our observed depletion profile in our CD169-DTR mice being restricted to only a subset of this F4/80^{hi} fraction.

Unfortunately, a multiparameter-flow cytometry would not help much in the characterization of these two fractions since, as shown in the heat-map below, the two cell types (clusters 0 and 1) show high similarities.

We can include these data as a supplementary figure if requested by the reviewer.

4) If the fungal burden is the same in WT and CD169 DTR mice at day 6, why are there differences in kidney damage (tubular necrosis, hemorrhages, etc)?

The increased in renal damage at day 6 was likely due to the heightened inflammation as shown by the increased TNF α level in the CD169-DTR kidneys [Figure 3E]. CD169⁺ macrophages have previously been shown to be anti-inflammatory, wherein absence of these macrophages led to over inflammation in bacterial infection or IRI injuries (Svedova et al., 2017, Karasawa et al., 2015).

This statement has been added to the 4th paragraph of the Result section - "*CD169-DTR mice suffered from irreversible, progressive renal damage during Candida infection*"

5) Are all of the C albicans alive in both groups? This should be addressed.

Yes, they were alive in both groups. As shown in Figure 2, fungal burdens from both groups were quantified by counting the colonies formed after 48 hours of incubation.

6) The data presented in figure 5I claiming that "Exogenous rIFN γ rescued neutrophils' oxidative killing in infected CD169-DTR mice" is a bit of a stretch as IFN γ treatment only partially rescues ROS production. Does treatment with IFN γ change the % of neutrophils that produce ROS? This would help to convince me that IFN γ does indeed stimulate ROS production in neutrophils.

Yes, IFN γ treatment does induce higher % of ROS-producing neutrophils. We have included this data in Figure 6D and F.

7) Why the immediate focus on NK cells in figure 6? Several cells can produce IFN γ including Th1 cells, ILC1s, NKT1s, Nk cells, and macrophages. It is unclear as to why the authors jump to exclude NK cells as being important without first identifying the major cell type that produces IFN γ , which they show are kappa⁺ B cells in the last figure. This figure is confusing and should be removed (or at the very least, moved to supplementary).

Indeed, the major or classical IFN γ producers have been known to be NK, NKT and T cells. As we assessed the expression level of IFN γ at the early stage of infection (Day 6 and 10), it was likely that T cells were not the major players considering that the adaptive immune response will take 2 to 3 weeks to set in (based on our vaccination experiments). Additionally, the involvement of NK cells in regulating the innate *Candida* immunity has been illustrated in several papers in recent years (explained in greater details in Results session - *NK cells were not the major IFN γ producers for renal Candida infection* and Discussion session - *7th paragraph*). In other words, on day 2 and 6 of the infection, immunity is dominated by innate responses of which NK cells have been studied extensively for their capacity to produce IFN γ in *Candida* immunity, hence we were interested to investigate whether NK cells were important in our experimental model. However, much to our surprise, NK cells appeared to be redundant as shown in Figure 6F. Hence, we attempted to investigate the types of cells that express IFN γ by doing FACS. Based on our FACS data, these IFN γ -producers were not CD3⁺ and CD49⁺, hence ruling out T, NK and NKT cells. Surprisingly, it was a unique population that appears to express B-cell restricted markers, CD19 and Kappa light chain (innate B-like cells).

8) In my opinion, the authors should make figure 5 G-J a separate figure. Figure 5 will show that ROS production is defective in CD169 deficient mice. New figure 6 would show that an inducer of ROS, IFN γ , is also defective and that treatment with IFN γ partially rescues ROS production.

We have made a separate figure for Fig. 5G-J as suggested to improve clarity.

It is my opinion that the authors should then show the major cell type that is producing IFN γ , which are kappa⁺ B cells. The major experiments that are needed to -decisively show the mechanism are:
1) to show that B cell specific IFN γ

production is lost in CD169⁺⁺ DTR mice and 2) to show that loss of kappa⁺ B cells (or IFN γ produced by these cells) mimics the phenotype found in CD169⁺⁺ deficient mice. However, this may be beyond the scope of this manuscript.

As shown in Figure 8B, we compared the number of IFN γ ⁺ B cells in uninfected control, infected WT and CD169-DTR mice and showed that the number of IFN γ -producing B cells was reduced in CD169-DTR mice.

As pointed out by the reviewer, the detailed characterization of these IFN γ -producing B cells is beyond the scope of this manuscript.

9) The section titled "Fraction I and Fraction II populations displayed differential expression profiles." comes out of nowhere and doesn't fit with the story. Why do the authors show expression of CX3CR1, Il12p35/p40 and Il10 here? If the authors are trying to characterize fraction 1 and fraction 2 macrophages from a wild type mouse, as they should, they should include this data earlier. RNA sequencing or multiparameter flow cytometry of fraction 1 and fraction 2 resident macrophages would be much more informative. This goes with the comment from above.

We attempted to characterize both fractions of renal macrophages and it appears that although they are very similar, only a subset of these macrophages expresses CD169. For more details, please refer to the answers for Reviewer 1, Qn3.

We have removed this section as suggested as it does seem to be a misfit to the story.

10) Can CD169⁺ macrophages directly influence ROS production by neutrophils and is this dependent on IFN γ ? This is especially relevant since IFN γ treatment only mildly rescued ROS production in neutrophils in figure 6A.

Adoptive transfer of CD169⁺ macrophages at the kidney may address this question but unfortunately, we do not have this data. In Figure 6C and D, we showed that exogenous IFN γ rescues neutrophils' ROS production and increases ROS-producing neutrophils. However, in Figure 8, we showed that IFN γ is not produced by macrophages, hence indicating that CD169⁺ macrophages indirectly influence the ROS level of the neutrophils.

Next, we sought to find the source of IFN γ and in Figure 8, we showed that IFN γ are produced by B-like cells. Hence, our data suggest that, together with B cells, CD169⁺ macrophages coordinate ROS secretion by neutrophils. Further research is warranted to investigate the efficiency of fungal killing by comparing the interplay between CD169⁺macrophages/neutrophils and CD169⁺macrophages/B-like cells/neutrophils.

Reviewer #2:

The authors provide convincing evidence that CD169⁺ macrophages are required for the defense against renal candidiasis in the mouse. This is an interesting finding worth reporting, since little is known about these macrophages in the kidney, on their role in the defense against kidney infection and on a role against candidiasis.

There are however a number of questions that need to be addressed before acceptance can be recommended. Most notably, the mechanistic clarification lacks depth.

Major concerns

1) the authors should characterize their depletion model more carefully. They only showed how many macrophages defined by different F4/80 and CD11b expression levels were depleted. First, they should show how efficiently CD169⁺ cells were depleted in the kidney and other organs. Next, it is essential to show selectivity of depletion, because if other cell types are depleted as well, conclusions on macrophages are not justified any more. This should be done under homeostatic conditions and infection, where CD169 expression might differ, and other cell types might become susceptible to depletion.

The CD169-DTR mouse used was generated in our laboratory as described in Purnama et al 2014. We have since assessed the depletion profiles of our CD169-DTR mice in different organs (Gupta et al., 2016, Purnama et al., 2014, Chen and Ruedl, 2020) and observed specific and selective depletion of tissue-resident macrophages in e.g. spleen, lungs, adipose tissue, liver and kidney. Over the past years, our studies showed that Ly6C^{hi} monocytes, monocyte-derived macrophages as well as dendritic cells are not affected upon DT treatment, except for a minor cross-priming M ϕ /cDC2 “hybrid” F4/80^{hi} antigen-presenting cell subset which express CD169 and therefore also depleted in CD169-DTR mice (Sheng et al. 2017)

We also monitored cell ablation profiles in kidney before and after *Candida* infection and have not observed susceptibility of other cell types to DT depletion [Figure 4a]. Below are some other cell types that we investigated.

We have added this information in the section - “CD169⁺ macrophages are a subpopulation of renal tissue-resident macrophages”

2) Previous studies by others have located CD169+ cells in the vicinity of renal vessels. Fig 1C is intended to show their location in the present study. However, this immunofluorescence microscopy shows no topological information. The authors should use immunohistochemistry.

In this paper, we were mainly interested in the gross location of CD169⁺ macrophages in the kidneys, whether they are located in the cortical (glomeruli) and/or the medullar region (renal tubules). For more details, please refer to our response to Reviewer 1, Qn 1.

3) the authors only used wild type mice injected with diphtheria toxin as controls. The control of PBS injected 169-DTR mice is lacking. This does not need to be repeated for all experiments, it is sufficient to show in one experiment that this transgenic mouse line is not more susceptible to candida than wild type mice

Indeed, we are aware that this is an important control to ensure that the phenotype observed in DT-treated CD169-DTR mice is not due to the effect of DT administration and has previously conducted a preliminary experiment similar to what is suggested, by comparing WT+DT mice with CD169-DTR+PBS mice. Our data revealed that their survivals were comparable.

4) it remains unclear how CD169+ macrophages contribute to the defense against candida. The authors speculate that an observed reduction in IFN γ is responsible, and identify not further characterised immune cells as potential producers. First, they did not provide evidence that the lack of IFN γ is causal. The authors show that other inflammatory cytokines are reduced as well (Fig S3). Why then IFN γ , and not these other cytokines?

The expression of IL12 (IL12p35/IL12p40) leads to IFN γ production. IFN γ was used as the target since IFN γ was the only cytokine investigated that showed the most reduction in CD169-DTR mice. As mentioned in the discussion (paragraph 6), we have also attempted to investigate other cytokines that have been known to be important for systemic *Candida* infection, such as IL6 and IL17. However, their qPCR expression level was undetectable.

We agreed that the inclusion of S3 is a misfit as suggested by Reviewer 1, Qn 9 and have removed it.

5) And even if IFN γ was causal, the connection to CD169⁺ macrophages is unclear. They did not show that the depletion of CD169⁺ macrophages reduces their IFN γ production. They also did not identify the molecular mechanism by which macrophages induce IFN production in these unidentified cells, but that is probably be beyond the scope of the present study.

We showed that there was a prominent increase in IFN γ expression (in qPCR and FACS) in infected WT mice, and this IFN γ expression was significantly reduced in infected CD169-DTR mice [Figure 6A, B]. In addition, absence of CD169⁺⁺ macrophages led to a significant reduction of IFN γ -producing B-like cells [Figure 8B]. Indeed, we acknowledge that current data have yet established any direct correlation between CD169⁺ macrophages and IFN γ -secreting B-like cells. However, it is clear that CD169⁺ macrophages do not contribute to the IFN level in kidney, which is the key driver for neutrophils' ROS production.

Minor concerns

- it should be mentioned that human kidneys are not as susceptible to candidiasis as mouse kidneys. Serious kidney candida infection is only seen in immunocompromised patients

This statement has been added to the last few sentences of 2nd paragraph in the discussion.

- How much were Cd169+ cells reduced in CX3CR1 Kos?

The reduction of CD169⁺ macrophages in CX3CR1-ko mice was similar to those in CD169-DTR mice [Supplementary Figure 1A and B]. Specifically, the average numbers of CD169⁺ macrophages in Control, CD169-DTR and CX3CR1-ko mice were 58,440, 4455 and 4986 respectively.

- Fig 4: Monocytes were defined as Ly6C⁺ CD11b⁺ cells but that includes also neutrophils

Neutrophils were excluded when we analyzed for the monocytes. Below is our gating strategy for monocytes and neutrophils. Indeed, for better clarity, we will change our definition of monocytes to Ly6C^{hi} CD11b⁺ cells.

Reviewer #3:

In this manuscript by Ruedl and colleagues, the authors have investigated the roles of CD169-expressing cells in invasive candidiasis using a CD169-DTR mouse. The authors describe that the mice lacking the CD169-expressing cell-types are susceptible to invasive candidiasis and display enhanced kidney damage and inflammation. Based on the data, the authors surmise that the CD169-expressing cells regulate IFN-g production by B-cell like cells and in turn regulate the PMNs' candidacidal activity. Although the role of CD169-expressing cells during invasive candidiasis has not been characterized before and therefore the work of potential interest, there are several concerns regarding the data and the authors' conclusions that are not fully supported by the findings. I have outlined my major concerns below:

1) CD169 has been shown to be expressed on a subset of renal tissue resident macrophage, hence the rationale for using a CD169-DTR mouse to assess this macrophage population is meritorious. In addition to the above resident macrophages, monocytes have also been shown to express CD169 (PMID: 25266072). Hence, with the authors' approach, DT administration should ablate these monocytes as well. However, based on their data this does not seem to be the case. The authors should address this discrepancy. Additionally, it would be ideal if the authors provide a portrait of different cell-types which express CD169 during invasive candidiasis using FACS. This will help the readers understand the target cells of the DT depletion of CD169 in their model.

This paper (PMID: 25266072) shows that in the adoptive transfer experiment, Ly6C^{lo} monocytes improve the survival rate of CD169-DTR mice by 62.5% and Ly6C^{hi} monocytes improve by 33%, which suggests that Ly6C^{lo} monocytes replace CD169⁺ macrophages. Additionally, using parabiosis experiment, the authors revealed that a small population of CD169⁺ macrophages is differentiated from Ly6C^{lo} monocytes. Approximately 5.5% of monocytes (CD11b⁺F4/80neg) are labelled CD169-YFP. In view of this, we would expect a partial depletion of Ly6C^{lo} monocytes in our CD169-DTR mice. Interestingly, in one of our earlier depletion experiments, as the figure (below) shows, there were no significant reduction of both Ly6C^{lo} and Ly6C^{hi} monocytes in our transgenic mice.

The presence of a small subset of CD169⁺Ly6C^{lo} monocytes was mentioned to the 4th paragraph of the Discussion section.

Noteworthy, the CD169-DTR mice (PMID: 25266072 and ours) have been generated from 2 different labs. Our CD169-DTR mice were generated in-house (Purnama et al., 2014). Therefore, one plausible explanation for the discrepancies in the deletion of monocytes in PMID: 25266072 and our model were because of the different CD169-DTR mice strains used.

2) The authors surmise that CD169⁺⁺ macrophages regulate PMNs' candidacidal activity. This conclusion is solely based on their observation that the PMNs from CD169-DTR mice have only slightly lower ROS production capacity. Hence, impaired candidacidal activity is indirectly inferred from the decreased ROS production capacity. The direct *Candida* killing has not been examined at all. Several approaches to examine *Candida* killing has been described (PMID: 26791948; 32554707; other papers). The authors can examine PMNs-mediated *Candida* killing to directly support their claims. As of now, the claim that PMN candidacidal activity is impaired is not supported by the data.

We have previously attempted to assess the PMN's candidacidal activity (WT and CD169-DTR) *ex vivo*, however, we are technically not able to perform the experiment protocol.

3) *In vitro* studies have pointed towards protective roles of IFN- γ , however IFN- γ in invasive candidiasis has not been shown to be protective in conventional non-gnotobiotic animals (PMID: 9125557). In addition, patients with IFN- γ signaling defect do not develop invasive candidiasis, they are at risk for mycobacterial and endemic fungal infections. Hence, the authors' rationale of assessing IFN- γ deficiency as sole driver of susceptibility in CD169-DTR needs clarification and importantly, a decrease in IFN- γ in CD169-depleter mice is unlikely to explain the dramatic increase in mortality in the model. Do IFN- γ KO mice in the authors' hands phenocopy CD169-depleter mice? Can they rescue the phenotype of the CD169-depleter mice with recombinant IFN- γ ? Other mechanisms beyond IFN- γ are likely operational to explain the susceptibility of CD169-depleter mice in the model.

Our data suggest two factors that contribute to the immunopathology of CD169-DTR mice. One factor is the lack of IFN γ -mediated immunity and the other is over-inflammation. CD169⁺ macrophages have previously been shown to be anti-inflammatory, wherein absence of these macrophages led to over inflammation in bacterial infection or IRI injuries (Svedova et al., 2017, Karasawa et al., 2015). Correspondingly, we observed that there was a significant increase of TNF α in the kidneys of CD169-DTR mice [Figure 3E].

Notably, the increased inflammation in the kidneys could be due to the inability of CD169-DTR kidneys in containing the fungal growth [Figure 2B and C], hence indicating the reduced ability of fungal killing by neutrophils. To illustrate the involvement of IFN γ on neutrophil functions in CD169-DTR mice, we treated the latter with rIFN γ , and showed that rIFN γ increases ROS-producing neutrophils in the kidneys. We have included this data in Figure 6D and F.

4. In the final figures, the authors analyze IFN- γ cellular sources and conclude that a B cell like population is responsible for IFN- γ production. In order to support their conclusion, the authors need substantial amount of evidence and work. Some suggestions

as below:

a. Independent assessment of IFN-g production capacity by this specific cell-type in context of invasive candidiasis to rule out technical artifact (autofluorescence; fluorochrome overlap), which seems more likely than a true finding that a B cell like cell is the predominant IFN-g cell in the Candida-infected kidney. In fact, there are not many B cells in the Candida-infected kidneys further raising concern that the cell that the authors claim is a B cell is likely to be a different cell that flow cytometrically appears as if it expresses B cell markers due to artifact.

We have made several attempts to reduce the likelihood of technical artifact. These include incubating the cells with FC-block (CD16/32 antibody) that precede staining with antibodies for extracellular markers, dead cell exclusion to avoid false-positive signals and tested with multiple fluorochromes of same extracellular marker. To ensure that these were unique B cells, we revealed that these IFN γ -producing cells were only restricted to Kappa-light chain and CD19⁺ cells [Figure 7]. We did not exclude any IFN γ ⁺ cells in our FACS plots as illustrated in Figure 8.

As discussed in the discussion paragraph 8, the presence of this unique B cell population in innate immunity has been reported in different infection models (Bao et al., 2014, Krocova et al., 2020).

To further investigate the presence of IFN γ ⁺ B cells in steady state kidney, we analysed scRNAseq data that was recently published by Zimmerman et. al. (Zimmerman et al., 2019). Here, we show that despite the low frequency of IFN γ ⁺ B cells at steady state, their presence in another study further substantiate the presence and expansion of such cells in the infection context.

b. In order to assess CD169⁺⁺ cell dependence of IFN- γ production by this specific cell population, the authors need to show whether in the CD169-DTR mice, this cell population has impaired IFN- γ production.

We have performed the experiment where we compared the number of IFN γ ⁺ B cells in uninfected control, infected WT and CD169-DTR mice. In this figure, we showed that IFN γ -producing B cells were indeed lost in CD169-DTR mice [Figure 8B].

Additionally, there are some other minor issues which also require further attention. I have highlighted them below:

1. The representative flow cytometry plot for CD169-DTR mice (and also control mice) have been used at multiple locations throughout the manuscript, this should not be the case. A same flow cytometry plot cannot represent independent experiments.

Thanks for raising it; it was a careless mistake from our side. We have changed the FACs plot.

2. From Figure 1, the authors mention that the CD169⁺⁺ macrophages are present in the medullary region and are ablated in the CD169-DTR mice. However, based on their high magnification image, this is not clear. The authors should provide a larger view of the kidney where cortical/medullary/pelvic regions are clearly visible and thus the localization of the corresponding cell ablation can be visualized.

We have assessed different region of the kidneys and have observed that CD169⁺ macrophages reside mainly in the medullary region. For more details, please refer to our response to Reviewer 1, Qn1.

3. In Figure 1A, the FACS ancestry of the gated F4/80-CD11b cells is not clear. It should be noted on which population these cells are gated on. It would also help if "gating strategy" for their flow cytometry experiments is provided in supplement.

This was our gating strategy:

4. In Figure 1D and in supplements, the Fraction I and Fraction II were sorted and utilized in experiments. In the supplement, the "post-sort" plots should be provided. This would be especially essential, if the cells are being sorted based on the gates' geometry as depicted in Figure 1A, since the 2 gates are pretty close to each other. The gates' proximity can cause cross-contamination in each fraction. Without the availability of "sorting strategy" and "post-sort" it is difficult to assess the purity of each macrophage fraction.

Although the two fractions are quite close, our sorting re-analysis clearly show two distinct purified cell fractions (see FACS plots below), when reanalyzed post-sort.

References for the responses to Reviewers' comments

- BAO, Y., LIU, X., HAN, C., XU, S., XIE, B., ZHANG, Q., GU, Y., HOU, J., QIAN, L., QIAN, C., HAN, H. & CAO, X. 2014. Identification of IFN-gamma-producing innate B cells. *Cell Res*, 24, 161-76.
- CHEN, Q. & RUEDL, C. 2020. Obesity retunes turnover kinetics of tissue-resident macrophages in fat. *Journal of Leukocyte Biology*, 107, 773-782.
- GUPTA, P., LAI, S. M., SHENG, J., TETLAK, P., BALACHANDER, A., CLASER, C., RENIA, L., KARJALAINEN, K. & RUEDL, C. 2016. Tissue-Resident CD169(+) Macrophages Form a Crucial Front Line against Plasmodium Infection. *Cell Rep*, 16, 1749-61.
- KARASAWA, K., ASANO, K., MORIYAMA, S., USHIKI, M., MONYA, M., IIDA, M., KUBOKI, E., YAGITA, H., UCHIDA, K., NITTA, K. & TANAKA, M. 2015. Vascular-resident CD169-positive monocytes and macrophages control neutrophil accumulation in the kidney with ischemia-reperfusion injury. *J Am Soc Nephrol*, 26, 896-906.
- KROCOVA, Z., PLZAKOVA, L., PAVKOVA, I., KUBELKOVA, K., MACELA, A., OZANIC, M., MARECIC, V., MIHELICIC, M. & SANTIC, M. 2020. The role of B cells in an early immune response to *Mycobacterium bovis*. *Microb Pathog*, 140, 103937.
- PURNAMA, C., NG, S. L., TETLAK, P., SETIAGANI, Y. A., KANDASAMY, M., BAALASUBRAMANIAN, S., KARJALAINEN, K. & RUEDL, C. 2014. Transient ablation of alveolar macrophages leads to massive pathology of influenza infection without affecting cellular adaptive immunity. *Eur J Immunol*, 44, 2003-12.
- SVEDOVA, J., MÉNORET, A., YEUNG, S. T., TANAKA, M., KHANNA, K. M. & VELLA, A. T. 2017. CD169+ Macrophages Restrain Systemic Inflammation Induced by *Staphylococcus aureus* Enterotoxin A Lung Response. *ImmunoHorizons*, 1, 213-222.
- ZIMMERMAN, K. A., BENTLEY, M. R., LEVER, J. M., LI, Z., CROSSMAN, D. K., SONG, C. J., LIU, S., CROWLEY, M. R., GEORGE, J. F., MRUG, M. & YODER, B. K. 2019. Single-Cell RNA Sequencing Identifies Candidate Renal Resident Macrophage Gene Expression Signatures across Species. *J Am Soc Nephrol*, 30, 767-781.

December 23, 2020

Re: Life Science Alliance manuscript #LSA-2020-00890-TR

Prof. Christiane Ruedl
Nanyang Technological University
School of Biological Sciences
60 Nanyang Drive
Singapore 637551
Singapore

Dear Dr. Ruedl,

Thank you for submitting your revised manuscript entitled "Renal CD169++ resident macrophages are crucial for protection against acute systemic candidiasis" to Life Science Alliance. The manuscript has been seen by the original reviewers whose comments are appended below. While the reviewers continue to be overall positive about the work in terms of its suitability for Life Science Alliance, some important issues remain.

As you will note from the reviewers' comments below, they are disappointed that their concerns about mechanistic studies were not sufficiently addressed. We are, however, sympathetic to your situation given that you have had to downsize your mouse colonies due to COVID-related lab shutdowns. We propose that you revise the manuscript in accordance to Reviewer 1's suggestions, leaving the mechanistic underpinnings for a future study. We would also appreciate it if you can incorporate Rev 2's and Rev 3's concerns in the discussion, and a point-by-point response to all the reviewers' concerns.

Please note that I will expect to make a final decision without additional reviewer input upon resubmission.

Please submit the final revision within one month, along with a letter that includes a point by point response to the remaining reviewer comments.

- A letter addressing the reviewers' comments point by point.
- An editable version of the final text (.DOC or .DOCX) is needed for copyediting (no PDFs).
- High-resolution figure, supplementary figure and video files uploaded as individual files: See our

detailed guidelines for preparing your production-ready images, <https://www.life-science-alliance.org/authors>

B. MANUSCRIPT ORGANIZATION AND FORMATTING:

Sincerely,

Shachi Bhatt, Ph.D.
Executive Editor
Life Science Alliance
<https://www.lsjournal.org/>
Tweet @SciBhatt @LSAJournal

Reviewer #1 (Comments to the Authors (Required)):

The first portion of this manuscript (Figures 1-4; supplemental figures S1-S3) describing the differential contribution of kidney resident macrophage subsets to *C. albicans* infection is strong, novel, and interesting. However, the remaining portion of the manuscript is disorganized, illogical, and speculative. For example, the authors propose that CD169 DTR mice have increased fungal burden and tissue damage due to the impaired ability of neutrophils to produce ROS. However, they never actually confirm that ROS production (overall ROS or neutrophil specific ROS) has any effect on fungal burden. This is a crucial mechanistic experiment that is missing if they are going to make this claim. Likewise, they propose that reduced B cell derived Ifng in CD169 DTR mice causes the reduced ROS production in neutrophils resulting in increased fungal burden. However, the authors do not provide direct evidence that altering Ifng levels in B cells (or depleting B cells altogether) has any effect on neutrophil ROS production or fungal burden. Why would it not affect ROS production in monocytes, who's ROS production is also dampened in CD169 DTR mice? Overall, the mechanistic link connecting Fr I macrophages, altered Ifng production in B cells, and altered ROS production in neutrophils is sorely lacking. I also do not understand why the authors could not perform a multi-parameter flow cytometry experiment to characterize Fr I and Fr II macrophages? Nevertheless, this reviewer is still enthusiastic about the first 4-5 figures of the paper despite the significant limitations of the remaining figures. This reviewer strongly recommends that the authors remove Figures 6, 7, and 8; however, it may be reasonable to include data from figure 5, which would allow them to speculate that the increased fungal burden in CD169DTR mice is due to the inability of neutrophils to produce ROS. The authors could then focus their future studies on developing a more detailed mechanism describing how Fr1 macrophages control fungal burden through interaction with IFNg+ B cells and ROS+ neutrophils. These changes, as well as the corresponding

changes to the text, would render this manuscript acceptable for publication in my opinion (even in the absence of a better characterization of CD169⁺⁺ and CD169⁺⁺⁺ macrophages). I have a few other responses to the authors comments below.

3) This paper would be significantly strengthened if a more thorough characterization (multiparameter flow; RNA seq) of CD169⁺⁺ and CD169⁺ resident macrophages was performed. This includes a better analysis of how much CD169 is expressed in individual fraction 1 and fraction 2 resident macrophages through multi-parameter flow cytometry.

To confirm the presence of two main F4/80^{hi} TRM subpopulations (Fraction I and II), we have analysed scRNAseq data that was recently published by Zimmerman et al. (Zimmerman et al., 2019). Among other identified CD45⁺ immune cells (NK-cells, T cells, ILC, B cells and neutrophils), the UMAP analysis visualizes a large cell fraction of F4/80⁺ macrophages which can be subdivided in two main (0 and 1) and one minor (3) subset. Of note, a weak Siglec-1 (CD169) signal is only detectable in Fraction 0 which indicates that only one of these F4/80⁺ subsets express detectable levels of this I-type lectin, hence supporting our observed depletion profile in our CD169-DTR mice being restricted to only a subset of this F4/80^{hi} fraction.

Unfortunately, a multiparameter-flow cytometry would not help much in the characterization of these two fractions since, as shown in the heat-map below, the two cell types (clusters 0 and 1) show high similarities.

We can include these data as a supplementary figure if requested by the reviewer.

While it is possible that Fr I and Fr II macrophages differ only in expression of CD169, this is highly unlikely based on the data the authors present in the paper showing that the function of each macrophage population is unique. Thus, it is very likely that Fr I and FrII macrophages would be different if the authors analyzed a series of commonly accepted macrophage markers in both cell types. It is unclear why the authors did not do this experiment. Also, while attempting to identify CD169⁺ resident macrophages in the single cell data set, the authors must remember that the sensitivity of single cell RNA sequencing is not good, which precludes this from being useful in answering the proposed question.

7) Why the immediate focus on NK cells in figure 6? Several cells can produce IFN γ including Th1 cells, ILC1s, NKT1s, Nk cells, and macrophages. It is unclear as to why the authors jump to exclude NK cells as being important without first identifying the major cell type that produces IFN γ , which they show are kappa⁺ B cells in the last figure. This figure is confusing and should be removed (or at the very least, moved to supplementary).

Indeed, the major or classical IFN γ producers have been known to be NK, NKT and T cells. As we assessed the expression level of IFN γ at the early stage of infection (Day 6 and 10), it was likely that T cells were not the major players considering that the adaptive immune response will take 2 to 3 weeks to set in (based on our vaccination experiments). Additionally, the involvement of NK cells in regulating the innate *Candida* immunity has been illustrated in several papers in recent years (explained in greater details in Results session - NK cells were not the major IFN γ producers

for renal Candida infection and Discussion session - 7th paragraph). In other words, on day 2 and 6 of the infection, immunity is dominated by innate responses of which NK cells have been studied extensively for their capacity to produce IFN γ in Candida immunity, hence we were interested to investigate whether NK cells were important in our experimental model. However, much to our surprise, NK cells appeared to be redundant as shown in Figure 6F. Hence, we attempted to investigate the types of cells that express IFN γ by doing FACS. Based on our FACS data, these IFN γ -producers were not CD3 $^+$ and CD49 $^+$, hence ruling out T, NK and NKT cells. Surprisingly, it was a unique population that appears to express B-cell restricted markers, CD19 and Kappa light chain (innate B-like cells)

The authors have missed the point of my original comment. Although there may be validity in hypothesizing that NK cells are important in controlling infection through production of IFN γ (based on literature), why not directly test if they produce IFN γ before going through all the work of depleting NK cells in their model? Also, if the authors claim that "it was likely that T cells were not the major players considering that the adaptive immune response will take 2 to 3 weeks to set in", why do they find IFN γ producing B cells in the kidney at days 2 and 6? B cells are adaptive immune cells, so this statement doesn't make sense and in fact, does suggest that the adaptive immune response can occur within 6 days post infection.

Reviewer #2 (Comments to the Authors (Required)):

The authors chose not to provide the requested analysis on the specificity and effectivity of their CD169 DTR mice in the candida infection model. They refer to previous publications, but these were not done in the candida system.

They also chose not to determine the numbers of renal CD169 $^+$ macrophages as requested. The request to improve the mechanistic clarification of their observation was mainly addressed by removing data and rewording, not by providing further mechanistic insight. My other points were addressed satisfactorily.

Reviewer #3 (Comments to the Authors (Required)):

In a revised manuscript, Ruedl and colleagues have addressed some of the concerns raised during the review process; some satisfactorily but some not. Although the characterization of CD169-expressing cells in renal antifungal immune response is meritorious and the revised manuscript is improved, there are three remaining items that, in the opinion of this reviewer, require further change:

1. The authors did not provide a comprehensive characterization of the different renal cell-types that express CD169 during invasive candidiasis. This can be easily done via flow cytometry using an anti-CD169 antibody which is commercially available. Based on such cell-type specific CD169 expression profiling, it will be easier for the readers to assess the CD169-DTR mice and its phenotypes in an unbiased manner and will provide important information on the molecule the authors' paper is about.
2. All reference to candidacidal activity of neutrophils or killing activity of neutrophils is not based on experimental data but inference from ROS differences. I understand it might be technically difficult

to do killing assays from neutrophils harvested from the kidney but the authors should remove all reference to killing by neutrophils as they did not perform such experiments. They should refer to ROS by neutrophils, not killing. Alternatively, they can examine neutrophil killing and refer to the results of that experimentation.

3. The authors provide some rationale about the cell type that expresses kappa light chain, intermediate levels of CD19, MHCII and produces IFN γ . It remains unclear what cell this is. A possibility, that is easily tested using flow cytometry is plasma cells. The authors should stain for CD138, BCMA and/or CD27 to rule in or rule out this CD19^{int} population is plasma cells. In addition, a figure that summarizes a) all IFN γ ⁺ cells in the infected kidney (how many are NK cells, how many this CD19⁺ population, how many are CD4⁺ T cells, how many CD8 T cells, how many NKT cells), and b) whether the frequency of the CD19⁺ population is decreased in CD169-DTR mice (which would go together with data requested in #1 above; that is, are these CD19⁺ cells expressing CD169) are important to show.

Dear Dr. Bhatt,

Thank you for giving us the opportunity to revise the manuscript and send back a final draft, which we hope will be accepted for publication in LSA.

We have addressed all points raised by the three reviewers and took out all our experiments related to the understanding of the mechanistic relationship between TRMs, ROS and IFN- γ . We still feel to keep the results showing the reduced renal IFN- γ response in absence of TRMs and the flow cytometry analysis of the putative IFN- γ secreting cells. We agree with you and the editors that new experiments addressing the mechanistic aspects between TRMs, neutrophils and IFN- γ producing cells are required and will be performed in a separate study.

See below our Point-by-Point reply to the latest reviewers comments.

Sincerely yours

Christiane Ruedl

Point-by-Point reply

We thank again all the reviewers for their constructive comments on our work. We have tried again to address as well as we can on the experimental concerns and provide detailed answers to the raised remarks. Amendments in the manuscript have been highlighted in red.

We hope that with the described changes and responses, the paper is now suitable for publication in LSA.

Reviewer #1:

While it is possible that Fr I and Fr II macrophages differ only in expression of CD169, this is highly unlikely based on the data the authors present in the paper showing that the function of each macrophage population is unique. Thus, it is very likely that Fr I and FrII macrophages would be different if the authors analyzed a series of commonly accepted macrophage markers in both cell types. It is unclear why the authors did not do this experiment. Also, while attempting to identify CD169+ resident macrophages in the single cell data set, the authors must remember that the sensitivity of single cell RNA sequencing is not good, which precludes this from being useful in answering the proposed question.

Reviewer #2:

The authors chose not to provide the requested analysis on the specificity and effectivity of their CD169 DTR mice in the candida infection model. They refer to previous publications, but these were not done in the candida system.

We agree that this manuscript lacks comprehensive profiling of Fr I and Fr II macrophages during the progression of the candida infection. We are currently not able to perform the experiment requested since unfortunately due to the pandemic we lack the mouse numbers requested for this experiment.

The CD169-DTR mouse model was generated many years ago in our lab and used in many other studies involving viral as well as parasite infections. All our data and tissue analysis generated in these studies have demonstrated a TRM specific ablation in all organs tested. Here in this manuscript we have shown that CD169-DTR mouse strain is a specific tool to ablate renal CD169⁺⁺ TRM while sparing neutrophils, monocytes and CD169⁺ TRM. We have also shown that CD169-DTR does not negatively impact other cell types investigated, such as T cells and NK cells. For ablation effectiveness, our data showed that the depletion of CD169⁺⁺ TRM is >90% [Fig. 1A, B].

They also chose not to determine the numbers of renal CD169+ macrophages as requested.

We have previously provided the number of CD169⁺ macrophages as requested. This is what was replied (quoted from previous point-by-point reply)

"• How much were Cd169+ cells reduced in CX3CR1 Kos?

The reduction of CD169⁺ macrophages in CX3CR1-ko mice was similar to those in CD169-DTR mice [Supplementary Figure 1A and B]. Specifically, the average numbers of CD169⁺ macrophages in Control, CD169-DTR and CX3CR1-ko mice were 58,440, 4455 and 4986 respectively. "

The request to improve the mechanistic clarification of their observation was mainly addressed by removing data and rewording, not by providing further mechanistic insight.

We agree that this manuscript lacks conclusive data to shed the lights on mechanistic insights. In the discussion, we have included additional references to discuss the discrepancy and probable insights to our studies. The elucidation of the mechanistic link between TRMs, neutrophil and IFN γ will be addressed in future studies.

My other points were addressed satisfactorily.

Reviewer #3:

In a revised manuscript, Ruedl and colleagues have addressed some of the concerns raised during the review process; some satisfactorily but some not. Although the characterization of CD169-expressing cells in renal antifungal immune response is meritorious and the revised manuscript is improved, there are three remaining items that, in the opinion of this reviewer, require further change:

1. The authors did not provide a comprehensive characterization of the different renal cell-types that express CD169 during invasive candidiasis. This can be easily done via flow cytometry using an anti-CD169 antibody which is commercially available. Based on such cell-type specific CD169 expression profiling, it will be easier for the readers to assess the CD169-DTR mice and its phenotypes in an unbiased manner and will provide important information on the molecule the authors' paper is about.

Using the flow-cytometry approach, we used two antibody clones (REA197 and 3D6.112) against CD169 (i.e Siglec-1) to assess the expression of Siglec-1 by cells isolated from kidneys. However, we had little success to get a good staining profile. It is important to note that Siglec-1 is believed to express very little in the kidney (Adapted from <https://www.genecards.org/cgi-bin/carddisp.pl?gene=SIGLEC1>).

In contrast, histological analysis gave clearer and definite positive staining results, hence we decided to include these data in figure 1. The discrepancy between flow cytometry and histology results could be explained by the possibility that the Siglec 1 molecule is sensitive to the enzymes used during the tissue digestion, which does not affect the expression of HB-EGF (DTR). The DTR-DT system is highly sensitive and few molecules of HB-EGF expression on the cell surface are enough to kill the expressing cells. The cells sensitive to DT ablation are expressing Siglec-1, although REA197 and 3D6.112 antibodies are not able to identify the positive cells clearly by FACS.

2. All reference to candidacidal activity of neutrophils or killing activity of neutrophils is not based on experimental data but inference from ROS differences. I understand it might be technically difficult to do killing assays from neutrophils harvested from the kidney but the authors should remove all reference to killing by neutrophils as they did not perform such experiments. They should refer to ROS by neutrophils, not killing. Alternatively, they can examine neutrophil killing and refer to the results of that experimentation.

We agree that we should refer to ROS and not to neutrophils killing and the terms should not be used interchangeably, we have since amended the manuscript. We have included more details on why we previously referred ROS to neutrophils killing. Please refer to paragraph 6 of the discussion section.

3. The authors provide some rationale about the cell type that expresses kappa light chain, intermediate levels of CD19, MHCII and produces IFN γ . It remains unclear what cell this is. A possibility, that is easily tested using flow cytometry is plasma cells. The authors should stain for CD138, BCMA and/or CD27 to rule in or rule out this CD19^{int} population is plasma cells.

Yes, future experiments will be warranted to extensively assess this unique population, whether they are plasma cells or innate like cells. At present with the existing literature on hand, we would think that this population is likely to be innate cell.

In addition, a figure that summarizes a) all IFN γ ⁺ cells in the infected kidney (how many are NK cells, how many this CD19⁺ population, how many are CD4⁺ T cells, how many CD8 T cells, how many NKT cells), and b) whether the frequency of the CD19⁺ population is decreased in CD169-DTR mice (which would go together with data requested in #1 above; that is, are these CD19⁺ cells expressing CD169) are important to show.

We have previously shown various types of potential cells that may be the IFN γ -producers. Please refer to Fig. 7 (previously Fig.8). In this experiment, we gated on all IFN γ ⁺ cells and back-gated to the respective marker panels. This was how we derived that the IFN γ producing cells express B-cell marker phenotype. In addition, we also revealed the reduction of CD19⁺ IFN γ ⁺ cells in the absence of CD169⁺⁺ TRM [Fig.7B (previously Fig.8B)].

We have tested several macrophage-related markers (e.g. CSF1R, MertK, CD64) by multi-color flow cytometry and could not visualize major differences since both cell subsets are macrophages therefore expressing the same markers at similar levels. The scRNA analysis attached to our previous rebuttal shows that there is a distinction between the populations based on their gene expression profile, therefore distinct populations are clearly separated in the tNSE plot analysis proving evidence for the diversity of this F4/80^{hi} expressing macrophage population.

The authors have missed the point of my original comment. Although there may be validity in hypothesizing that NK cells are important in controlling infection through production of IFN γ (based on literature), why not directly test if they produce IFN γ before going through all the work of depleting NK cells in there model? Also, if the authors claim that "it was likely that T cells were not the major players considering that the adaptive immune response will take 2 to 3 weeks to set in", why do they find Ifng producing B cells in the kidney at days 2 and 6? B cells are adaptive immune cells, so this statement doesn't make sense and in fact, does suggest that the adaptive immune response can occur within 6 days post infection.

We have now rearranged the main figures and supplementary figures as well as the text according to reviewers' advice. In brief, we now show that a pilot experiment was first carried out to screen the cell source that produce IFN γ which includes surface marker to identify NK cells. After we observed no IFN γ secretion from NK cells, we followed up with a series of experiments to validate our observation. We included NK cells as it has been described as one of the major innate cells to release IFN γ and to regulate neutrophils' functions. It is surprising that NK cells are not vital for producing IFN γ in our infection model, hence we believe that it will be informative to share our observations with the community. Moreover, for those papers that we mentioned in the discussions, the function of NK cells was assessed at an earlier timepoint (i.e day 1), which could be the reason for the discrepancy observed. For more details, please refer to the 2nd last paragraph of the discussion section.

We would like to highlight that we also assessed on the possibility of T cells being the IFN γ producers. However, our data revealed that they do not express IFN γ at day 6 (previously in Fig 8; now in Fig 7). In this paper, we are proposing that it is a unique innate B-like cell that express IFN γ . It is vital to note that the strain that we used for infecting the mice, SC5314, is not a natural commensal for the mice; hence the likelihood of them developing adaptive immunity prior to infection is believed to be minimal.

January 19, 2021

RE: Life Science Alliance Manuscript #LSA-2020-00890-TRR

Prof. Christiane Ruedl
Nanyang Technological University
School of Biological Sciences
60 Nanyang Drive
Singapore 637551
Singapore

Dear Dr. Ruedl,

Thank you for submitting your revised manuscript entitled "Renal CD169++ resident macrophages are crucial for protection against acute systemic candidiasis". We would be happy to publish your paper in Life Science Alliance pending final revisions necessary to meet our formatting guidelines.

Along with the points listed below, please also attend to the following:

- please add a callout for Figure S3A to your main manuscript text
- please use the [10 author names, et al.] format in your references (i.e. limit the author names to the first 10)
- The uninfected panel in Fig S3A, B and C look like they are the same image. It is LSA policy to not duplicate any images within the manuscript. Given that the comparison is with CD169-DTR mice at day 3, 6 and 10 p.i, we would need you to provide appropriate experiment-matched controls within each figure.

A. FINAL FILES:

B. MANUSCRIPT ORGANIZATION AND FORMATTING:

Sincerely,

Shachi Bhatt, Ph.D.
Executive Editor
Life Science Alliance
<https://www.lsjournal.org/>
Tweet @SciBhatt @LSAJournal

January 22, 2021

RE: Life Science Alliance Manuscript #LSA-2020-00890-TRRR

Prof. Christiane Ruedl
Nanyang Technological University
School of Biological Sciences
60 Nanyang Drive
Singapore 637551
Singapore

Dear Dr. Ruedl,

Thank you for submitting your Research Article entitled "Renal CD169++ resident macrophages are crucial for protection against acute systemic candidiasis". It is a pleasure to let you know that your manuscript is now accepted for publication in Life Science Alliance. Congratulations on this interesting work.

DISTRIBUTION OF MATERIALS:

Again, congratulations on a very nice paper. I hope you found the review process to be constructive and are pleased with how the manuscript was handled editorially. We look forward to future exciting submissions from your lab.

Sincerely,

Shachi Bhatt, Ph.D.

Executive Editor

Life Science Alliance

<https://www.lsjournal.org/>
